# ReGraP-LLaVA:
# Reasoning enabled Graph-based Personalized Large Language and Vision Assistant

## Abstract

Multimodal Large Language Models (MLLMs) have demonstrated remarkable performance across a wide range of multimodal tasks. Recent advances in personalized MLLMs enable effective capture of user-specific concepts, supporting both recognition of personalized concepts and contextual captioning. However, humans typically explore and reason over relations among objects and individuals, transcending surface-level information to achieve more personalized and contextual understanding. To this end, existing methods may face three main limitations: (1) Their training data lacks multi-object sets in which relations among objects are learnable, (2) Existing models often neglect the connections between different personalized concepts, thereby failing to perform reasoning over them, (3) Their experiments mainly focus on a single personalized concept, where evaluations are limited to recognition and captioning tasks. To address the limitations, (i) We present a new dataset named ReGraP, consisting of 120 sets of personalized knowledge. Each set includes images, Knowledge Graphs (KGs), and Chain-of-Thought Question-Answering (CoT QA) pairs derived from the KGs, enabling more structured and sophisticated reasoning pathways. (ii) We propose **R**easoning **e**nabled **Gra**ph-based **P**ersonalized **L**arge **L**anguage **a**nd **V**ision **A**ssistant (**ReGraP-LLaVA**), an MLLM trained with the corresponding KGs and CoT QA pairs, where soft and/or hard graph prompting methods are designed to align KGs within the model's semantic space. (iii) We establish the ReGraP Benchmark, which contains diverse task types: Multiple-Choice, Fill-in-the-blank, True/False, and Descriptive questions in both open- and closed-ended settings. The proposed benchmark is designed to evaluate the relational reasoning and knowledge-connection capability of personalized MLLMs. We conduct experiments on the proposed ReGraP-LLaVA and other competitive MLLMs. Results show that the proposed model not only learns personalized knowledge but also performs relational reasoning in responses, achieving the best performance compared with the competitive methods. All the codes and datasets are released at: https://anonymous.4open.science/r/ReGraP.

## 1 Introduction

Achievements in MLLMs (Bai et al., 2023a; Liu et al., 2023a; Team et al., 2023; Zhu et al., 2024) have demonstrated robust capabilities in image analysis, and user prompts are employed to enable initial personalization for handling queries such as "*What is <Tom> doing in this picture?*" and "*Could you find <my headphone> in this image?*". Consequently, to advance beyond the constraints of user-prompt based personalization (Nguyen et al., 2024; Wu et al., 2024), recent approaches (Nguyen et al., 2024; Pham et al., 2024; Hao et al., 2024; Alaluf et al., 2024) introduce personalized concepts (e.g., a personal item or a specific individual) into models, yielding satisfactory performance on recognition and simple QA tasks. For example, Yo'LLaVA (Nguyen et al., 2024) introduces new tokens for representing personalized concepts to LLaVA (Liu et al., 2023a), achieving remarkable experimental results on personalized recognition and captioning while preserving the model's original conversational capabilities, which is the state-of-the-art approach in the personalization of MLLMs.

However, existing personalized MLLMs exhibit limitations. As illustrated in Figure 1 (a), prompt-based personalized MLLMs rely on user prompts as the sole source of personalized knowledge. This dependency leads to degraded performance when handling low-quality prompts and complicated

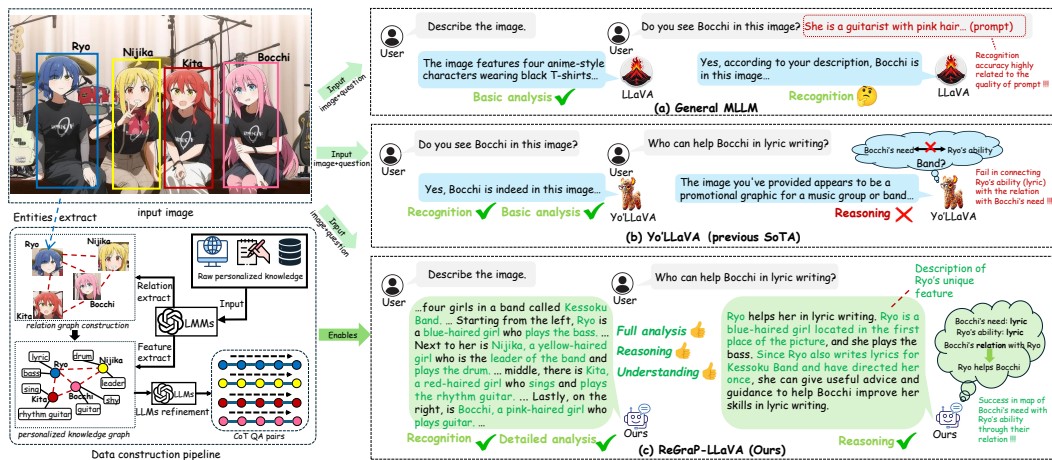

Figure 1: The comparison between ReGraP-LLaVA and other personalized MLLMs.

tasks (Nguyen et al., 2024; Wu et al., 2024). Meanwhile, training-based personalized MLLMs (e.g. Yo'LLaVA) in Figure 1 (b) mainly focus on concept-learning level, overlooking the human-like process of constructing a knowledge network among different items and leveraging the relations for contextual understanding (Gentner, 1983; Aminoff et al., 2022).

Consider the query: "*Who can help Bocchi in lyric writing?*". Addressing it requires models not only recognize people in the image, but also identify relations among different individuals and reason over their attributes and relations. Meanwhile, the combination of graphs with MLLMs can enhance models' performance (Wang et al., 2025; Hamza et al., 2025). To this end, constructing knowledge graphs (KGs) for personalized knowledge (personalized concepts, their attributes and relations) serves as a promising approach for training models to learn the relational knowledge. Besides, studies indicate that training on Chain-of-Thought (CoT) (Wei et al., 2022) data improves models' reasoning performance in image-related tasks (Xu et al., 2024; Yeo et al., 2025), which suggests that such data can be leveraged to enhance MLLMs' reasoning capability over personalized knowledge. Based on these insights, we raise three research questions:

- **RQ1**: Can we construct a dataset that integrates images, KGs, and CoT data to comprehensively encode personalized knowledge?
- **RQ2**: Given a dataset in **RQ1**, can we develop a personalized MLLM whose training framework aligns with the KGs, enabling it to learn and reason over the personalized knowledge?
- **RQ3**: Given a personalized MLLM in **RQ2**, can we evaluate its relational reasoning and knowledge connection capability, particularly for personalized queries that expect contextual responses?

In this paper, to address **RQ1**, we present a new dataset, ReGraP, consisting of 120 independent sets of personalized knowledge. The dataset is constructed through a data generation pipeline that builds KGs based on the images and personalized knowledge (see Figure 1 left) and subsequently derives Chain-of-Thoughts Question-Answering pairs (CoT QA pairs) from the KGs. The answers in CoT QA pairs incorporate comprehensive reasoning steps. To address **RQ2**, we propose ReGraP-LLaVA, a novel MLLM built on LLaVA and trained using the ReGraP dataset, incorporating images, CoT QA pairs, and KGs in its training framework. To align the graph-based structure of KGs with the token-based input paradigm of LLaVA, we transform the KGs into embeddings using Graph Neural Networks (GNNs) and projection modules which serves as a "soft-prompt" method, and convert KGs into sequences of relational descriptions and tokenize them through reasoning tokenizers by introducing new *entity tokens* and *relation tokens* which serves as a "hard prompt" method. ReGraP-LLaVA showcases the capability to capture personalized knowledge and utilize it for relational reasoning. To address **RQ3**, we establish the ReGraP benchmark to assess models' reasoning and knowledge-connection capabilities, rather than restricting the evaluation to basic recognition or general captioning tasks. This benchmark spans multiple-choice, fill-in-the-blank, true/false, and descriptive questions, covering both open- and closed-ended settings. Experimental results show that ReGraP-LLaVA achieves high performance on both basic tasks evaluating personalized knowledge acquisition and difficult tasks requiring relational reasoning over learned knowledge.

**Contributions.** In summary, our main contributions are:

- We present ReGraP dataset and the data generation pipeline for personalized MLLMs, containing knowledge graph construction and CoT QA pairs generation based on the constructed KGs.
- We propose ReGraP-LLaVA, a novel MLLM leverages soft and/or hard prompts of knowledge graphs and CoT QA pairs in training, and not only learns personalized concepts but also utilizes the relational knowledge among these concepts to perform reasoning, enabling comprehensive image analysis and question-answering.
- We establish the ReGraP benchmark, comprising Multiple-Choice, Fill-in-the-blank, True/False, and Descriptive questions across both open- and closed-ended settings. This benchmark scales in difficulty, measuring models' knowledge acquisition and relational reasoning capabilities.

## 2   RELATED WORK

**Multimodal Large Language Models.** Large Language Models (LLMs) (Achiam et al., 2023; Li et al., 2024; Weng et al., 2024; Liu et al., 2024a) have demonstrated remarkable capabilities in general question answering and reasoning. Building on this foundation, recent works have extended LLMs to visual domains, leading to the development of MLLMs (Bai et al., 2023a; Liu et al., 2023a; Team et al., 2023; Zhu et al., 2024; OpenAI, 2024; Bai et al., 2023b), which process both textual inputs and images, thus handling multimodal tasks. However, although these MLLMs possess extensive knowledge for handling general tasks (e.g., recognition and captioning), the lack of user-specific information limits their capabilities in handling personalized requests. In this work, we train MLLMs to learn personalized knowledge while preserving their original conversational capabilities.

**Personalizing MLLMs as AI Assistants.** User prompting is a direct and effective method to guide MLLMs aligning with users' preference with minimal cost, while its the performance heavily depends on the quality of user prompts and tends to degrade as task complexity increases. (Nguyen et al., 2024; Wu et al., 2024). Therefore, advanced approaches for personalization have been proposed, which can be categorized into retrieval based methods (Hao et al., 2024; Li et al., 2025; Das et al., 2025; Seifi et al., 2025) and model-training based methods (Nguyen et al., 2024; Pham et al., 2024; Alaluf et al., 2024; An et al., 2025). Retrieval based methods utilize a database to store the knowledge of personalized concepts. During inference, the system retrieves relevant information to determine whether the queried object corresponds to a personalized concept, and adapts its behavior accordingly across different scenarios. For example, RAP (Hao et al., 2024) presents a retrieval-augmented module that can be integrated into MLLMs, enabling scenario- and user-specific responses. However, these methods depend on external knowledge bases and additional models (e.g., YOLO (Redmon et al., 2016)) to extract personalized concepts. Model-training based methods introduce extra modules (Alaluf et al., 2024) and embeddings (Nguyen et al., 2024), training MLLMs to learn personalized concepts. These approaches rely solely on the MLLM itself. However, they typically limit personalization to object recognition and captioning, overlooking the relations among personalized concepts that can be structured into a knowledge graph to support relational reasoning. In contrast, our model captures personalized concepts, their attributes and relations, thus having relational reasoning capabilities and giving contextual responses when handling personalized queries.

## 3   REGRAP DATASET: DATA GENERATION PIPELINE

We introduce a data generation pipeline that provides KGs and CoT QA pairs for model training, as shown in Figure 2. Given a set of images and textual descriptions, this pipeline extracts personalized knowledge from the raw input and explores their relations, thereby constructing a personalized knowledge graph and subsequently generating CoT QA pairs from the constructed KG.

This pipeline aims at constructing a training database $\mathcal{D}$ consisting of multiple independent sets. Each set $\mathcal{S}$ represents data of a set of personalized concepts (e.g., individuals, items...) and their attributes and relations. It contains a collection of images $\mathcal{I}$ of these concepts, a KG $\mathcal{G}$ whose nodes are the concepts and their attributes, and edges are the relations, and a set of instruction pairs $\mathcal{P}_{(\mathcal{QA},\mathcal{R})}$ where each pair consists of a CoT QA pair $\mathcal{QA}$ and its corresponding reasoning subgraph $\mathcal{R} \subseteq \mathcal{G}$.

The images come from user-captured photos or publicly available sources on the internet, while textual knowledge $\mathcal{K}_{text}$ originates from users' own descriptions or publicly accessible resources. Images can directly serve as the $\mathcal{I}$. To this end, we focus on constructing $\mathcal{G}$ that fully captures both relational and attribute-based knowledge of $\mathcal{K}_{text}$, and subsequently utilize $\mathcal{G}$ to generate $\mathcal{QA}$. Accordingly, the data generate pipeline is divided to two main steps.

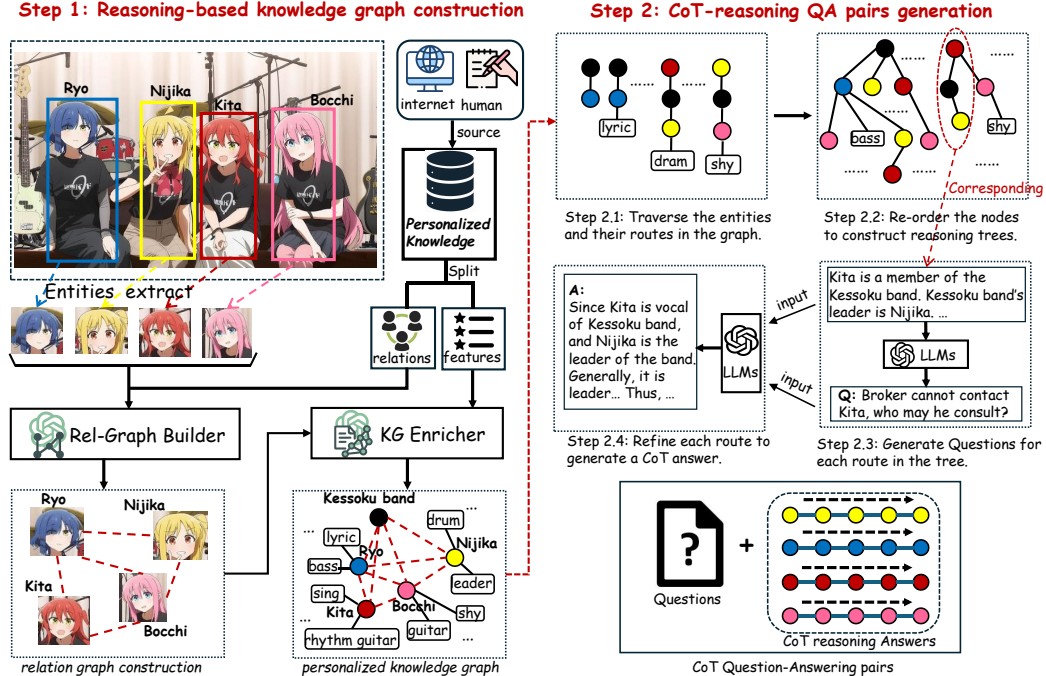

Figure 2: The data generation pipeline. We first construct knowledge graph that represents the personalized knowledge, and then derive CoT QA pairs from the knowledge graph.

**Reasoning-based Knowledge Graph Construction.** The construction of $\mathcal{G}$ is the first step of the data generation pipeline (see Figure 2 left). Personalized concepts in $\mathcal{I}$ are extracted as main entities $\mathcal{E}$, and $\mathcal{K}_{text}$ is divided into relational knowledge $\mathcal{K}_{\mathcal{R}}$ and attribute knowledge $\mathcal{K}_A$. We prompt GPT-4o (Hurst et al., 2024) to serve as a Relation-Graph Builder $\mathcal{B}_{RG}$ (prompts are detailed in Table 13). The builder takes $\mathcal{K}_{\mathcal{R}}$, $\mathcal{E}$, and $\mathcal{I}$ as input, and outputs a set of triplets that form the relation graph $\mathcal{G}_{\mathcal{R}}$ that contains the nodes of entities and edges of relations, formulated as:

$$\mathcal{G}_{\mathcal{R}} = \mathcal{B}_{RG}(\mathcal{K}_{\mathcal{R}}, \mathcal{E}, \mathcal{I}) = \{(h_i, r_i, t_i)\}_{i=1}^n, \tag{1}$$

where each triplet $(h_i, r_i, t_i)$ denotes a head entity $h_i \in \mathcal{E}$, a relation $r_i$, and a tail entity $t_i \in \mathcal{E}$, capturing the semantic connections derived from relational knowledge and image. Then, we prompt GPT-4o to serve as a KG Enricher $\text{Er}_{\mathcal{KG}}$ (prompts are detailed in Table 14), which takes $\mathcal{G}_{\mathcal{R}}$ and $\mathcal{K}_A$ as input. The process proceeds in two steps: (1) It adds nodes $N_{new}$ representing attributes and potential new concepts to the graph. (2) It explores and adds new edges representing relations between the nodes. This process generates the personalized knowledge graph $\mathcal{G}$, formulated as:

$$\mathcal{G} = \text{Er}_{\mathcal{KG}}(\mathcal{G}_{new}, \mathcal{K}_A), \text{with } \mathcal{G}_{new} = \mathcal{G}_{\mathcal{R}} + N_{new} \tag{2}$$

**CoT QA pairs Generation.** After constructing $\mathcal{G}$, we proceed to generate $\mathcal{P}_{(\mathcal{QA},\mathcal{R})}$ from the routes on $\mathcal{G}$, which is the second step of the data generation pipeline (see Figure 2 right). As the relations in $\mathcal{G}$ connect nodes to form step-by-step paths that collectively compose a reasoning chain, we begin by traversing the nodes and their relation paths to construct reasoning routes. In this context, the routes may share common starting nodes or be nested within longer routes. Therefore, we reorder the nodes to construct reasoning trees $\mathcal{T}$, which contains more comprehensive and hierarchically structured reasoning routes. Thereafter, Depth-first search (DFS) is applied to $\mathcal{T}$ to extract the longest reasoning routes (subgraphs) and construct a set of reasoning routes $\mathcal{S}_R$, formulated as: $\mathcal{S}_R = \text{DFS}(\mathcal{T})$. These routes are subsequently used as contextual prompts to guide GPT-4o in generating questions $\mathcal{Q}$, where each route $\mathcal{R}$ serves as the "thinking process" for answering the question. Finally, $\mathcal{Q}$ and $\mathcal{R}$ are jointly provided to GPT-4o to generate a CoT reasoning answer $\mathcal{A}_{CoT}$, formulated as:

$$\mathcal{A}_{CoT} = \text{GPT}(\mathcal{Q}, \mathcal{R}), \tag{3}$$

Each question and its answer form a CoT QA pair, associated with the corresponding $\mathcal{R}$. These pairs jointly constitute the $\mathcal{P}_{(\mathcal{QA},\mathcal{R})}$. Prompts for QA generation are detailed in Table 15 and Table 16. Section C evaluates the quality of the generated CoT QA pairs.

# 4 ReGraP-LLaVA: Training Framework

To align knowledge graphs with MLLMs, we adopt both soft prompts in Figure 3 (a) and hard prompts in Figure 3 (b) to transform the graph into a format compatible with LLaVA. In addition, the CoT QA pairs can be regarded as the extraction of knowledge in the graph, also serving as a hard-prompt formulation of knowledge graphs.

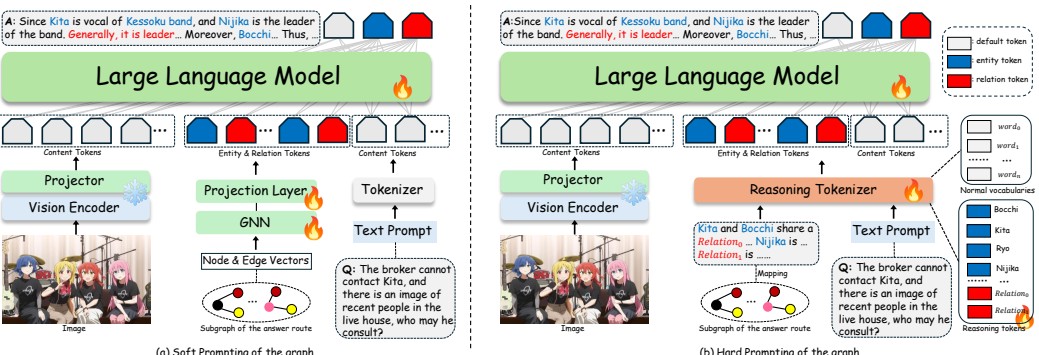

Figure 3: The framework of ReGraP-LLaVA. The left side shows the framework to soft prompt graphs, and the right side shows the framework to hard prompt graphs.

## 4.1 Soft-Prompting LLM with Graph.

This method employs a GNN module in conjunction with a projection layer, implemented as a multilayer perceptron (MLP), for encoding the subgraph into embeddings that are aligned with the vector space of the LLM. To prepare the graph for GNN processing, nodes and relations are first converted into one-hot encoded vectors, constructing a graph $g$, whose nodes are personalized concepts and their attributes, and edges are relations between the concepts and attributes. The graph $g$ is firstly passed through the GNN to compute the graph embedding $\mathcal{H}_g = \text{GNN}(g)$, which provides a representation of the personalized knowledge in the subgraph. To align the embedding with the same word embedding space of MLLM, we then apply a MLP projection layer to convert $\mathcal{H}_g$ to $\hat{\mathcal{H}}$ which has the same dimensionality as the word embedding space in the language model, formulated as:

$$\hat{\mathcal{H}} = \text{MLP}(\mathcal{H}_g) \in \mathbb{R}^d, \text{with } H_g = \text{GNN}(g) \tag{4}$$

where d is the hidden dimension of the LLM. $\hat{\mathcal{H}}$ is then used as a soft prompt in conjunction with embeddings derived from visual and textual inputs.

We proceed to process the textual and visional information associated with the subgraph. Specifically, the $\mathcal{Q}$ of the CoT QA pairs provide the language instruction, and the $\mathcal{A}_{CoT}$ serve as the Language Responses. In this framework, the vision encoder, vision projector, and tokenizer of the pre-trained LLaVA are frozen to preserve their original capabilities. The $\mathcal{Q}$ is first tokenized by the LLM's tokenizer into a sequence of discrete tokens $\{\text{token}_i\}_{i=1}^m$, where $m$ denotes the sequence length. Subsequently, these tokens are embedded into a continuous vector space, formulated as:

$$\mathbf{X}_{\text{emb}} = \text{Embed}(\{\text{token}_i\}_{i=1}^m) \in \mathbb{R}^{m \times d}, \tag{5}$$

Next, $\mathbf{X}_{\text{emb}}$ is concatenated with the graph embeddings $\hat{\mathcal{H}}$ to form the new instruction embedding token sequence $X_i^{\text{new}}$, denoted as $X_i^{\text{new}} = \mathbf{X}_{\text{emb}} + \hat{\mathcal{H}}$. Following the original LLaVA architecture, $X_i^{\text{new}}$ is concatenated with the visual embedding tokens encoded from the image input via the vision encoder and projector, and the sequence is processed following the original LLaVA training pipeline.

## 4.2 Hard-Prompting LLM with Graph.

While embedding a graph directly into vector representations aligns it with word embedding space, knowledge in a graph can also be represented by natural language, which also serve as learnable prompts for MLLMs. Inspired by recent studies (Nguyen et al., 2024; Pham et al., 2024) that learnable tokens can efficiently capture personalized concepts, we introduce new *reasoning tokens* to represent personalized knowledge. These tokens enable the subgraph to be expressed as a prompt sequence that integrates the newly added tokens, thereby allowing the model to learn the reasoning processes.

Specifically, a subgraph can be represented as a collection of continuous relational triples, formulated as $g = \{(E_i, r_i, E_{i+1})\}_{i=1}^N$, where each triple consists of a head entity $E_i$, a relation $r_i$, and a tail entity $E_{i+1}$. Here, $E_i$ denotes personalized concepts and their attributes, $r_i$ denotes relations, and $N$ denotes the total number of relations in the subgraph. The triples are sequentially connected. For entities, we utilize their names as the new-added *entity tokens* and $\{<\texttt{Relation}_i>\}_{i=1}^N$ as the new-added *relation tokens*. Hereby, the prompt to describe the graph is: "*<E₁> and <E₂> share a <Relation₁>,...,<E_N> and <E_{N+1}> share a <Relation_N>. <E₁> is <desc.>,...,<Relation₁> is <desc.>....*" Here, $E_i$ are the names of the entities, and *<desc.>* represents a short natural language description. Given a graph, there are $N$ *relation tokens* and $N + 1$ *entity tokens*, collectively referred to as *reasoning tokens*, which are learned to embed structural information of the graph. These reasoning tokens are introduced as new entries to the tokenizer, effectively transforming it into a reasoning tokenizer. Subsequently, the final classification head matrix $W_{c \times n}$ of the language model is expanded by $2N + 1$, resulting in an updated matrix $W_{c \times (n+2N+1)}$, to accommodate the additional vocabulary introduced by reasoning tokens. Thus, the new trainable parameters are:

$$\boldsymbol{\theta}_{new} = \{<E_1>, \ \ldots, \ <E_{N+1}>, \ <Relation_1>, \ \ldots, \ <Relation_N>, W_{(:,n+2N+1)}\}.$$

The $\mathcal{Q}$ of the CoT QA pairs and the graph prompt $X_g$ are concatenated to form new instruction input $X_i$, together with $\mathcal{A}_{CoT}$ and the associated image input $I$, constituting the training data triplets $(X_i, I, X_a)$ of LLaVA. We apply the standard language loss of LLaVA to compute the probability of the target answers $X_a$ for each conversation of length $L$:

$$p(\mathbf{X}_a \mid \mathbf{I}, \mathbf{X}_i) = \prod_{j=1}^{L} p_\theta \left( x_j \mid \mathbf{I}, \mathbf{X}_{i,<j}, \mathbf{X}_{a,<j} \right), \tag{6}$$

where $\theta$ denotes the trainable parameters of the model, and $\mathbf{X}_{i<j}$ and $\mathbf{X}_{a<j}$ represent the instruction and answer tokens from all previous turns prior to the current prediction token $x_j$, respectively.

## 5 EXPERIMENTAL SETUP

| Dataset | # Sets | Single Obj. | Multi Obj. | # Avg. | # Images/set | Text Desc. | CoT | Graph | Len. |
|---|---|---|---|---|---|---|---|---|---|
| MyVLM (Alaluf et al., 2024) | 30 | ✓ | ✗ | – | ~11.67 | ✓ | ✗ | ✗ | ~1 |
| Yo'LLaVA (Nguyen et al., 2024) | 40 | ✓ | ✗ | – | ~10 | ✓ | ✗ | ✗ | ~1 |
| ReGraP (Ours) | 120 | ✓ | ✓ | 5.5 | ~20 | ✓ | ✓ | ✓ | ~5.2 |

Table 1: Comparison between MyVLM (Alaluf et al., 2024), Yo'LLaVA (Nguyen et al., 2024) and ReGraP datasets. **Avg.**: avg. num. of objects in multi-obj. sets. **Len.**: avg. num. of steps in QA pairs.

**Training.** We have 10 training images and around 20 CoT QA pairs for one set of personalized knowledge. We use LoRA (Hu et al., 2022) and AdamW (Kingma & Ba, 2014) with a learning rate of 1e-5 and LLaVA-v1.6-vicuna-7b (Liu et al., 2023b) as the base model. We train each set for up to 10 epochs on single NVIDIA A6000.

**Dataset.** Building on the Yo'LLaVA dataset, we construct 80 additional personalized knowledge sets: 40 single-object, 20 five-object and 20 six-object, totally 120 sets. Images in the multi-object sets contain all personalized concepts of the corresponding set. In single-object sets, surrounding objects in images and textual knowledge provide supplementary entities to construct the knowledge graph. Each set contains approximately 10 training and 10 testing images, totally around 20 images. Additionally, each set contains a full graph, over 20 CoT QA pairs and corresponding subgraphs.

**Baselines.** For finetuning-based baselines, we select Yo'LLaVA (Nguyen et al., 2024) and LLaVA-7B (Liu et al., 2023b) with LoRA finetuning as our comparison models. Both are trained on raw personalized knowledge (images and the raw textual knowledge) and CoT QA pairs in ReGraP dataset separately, to demonstrate the effectiveness of both our ReGraP dataset and training framework.

For prompt-based baselines, we evaluate 7B models including LLaVA (Liu et al., 2023b), Qwen-2-VL (Wang et al., 2024) and Qwen2.5-VL (Team, 2025), 13B model LLaVA-1.5-13B (Liu et al., 2024b) and leading models Qwen2.5-VL-72B (team, 2025) and GPT-4o (Hurst et al., 2024) (GPT-4o is only evaluated in the close-ended tasks, as it is the evaluator in open-domain tasks). We prompt the models with descriptions of personalized knowledge in images, avoiding the leakage of direct answers.

**Benchmark.** For each set of personalized knowledge, The closed-ended questions contain 40 multiple-choice questions (20 basic, 20 requiring reasoning), 15 true-or-false questions (5 basic, 10

requiring reasoning), 10 fill-in-the-blank questions (5 basic, 5 requiring reasoning), and 3 descriptive questions (1 for basic captioning, 2 requiring reasoning). The open-ended questions contain 5 descriptive questions and 1 question asking models to generate a full description of the image.

# 6 RESULTS

We demonstrate the effectiveness of the ReGraP-LLaVA across close-ended question-answering in Section 6.1 and open-ended question-answering in Section 6.2. These tasks evaluate models' ability to recognize personalized concepts, learn and reason over their attributes and relations in various settings. Section 6.3 compares the performance between different graph-prompting methods. Section 6.4 shows qualitative examples of models' answers. Section A provides additional ablation studies, section B evaluates the alignment of models' responses with human's preference, and section I discusses possible errors and deviations. The closed-ended tasks with ground-truth answers has no reliance on GPT-4o or other external evaluators, thus provide the core objective results. The open-ended tasks complementarily illustrate the model's generative capabilities. Together with the human evaluation and the qualitative cases, the model's performance has been comprehensively evaluated, covering both objective fixed-answer benchmarks and subjective preference alignment.

| Model | Multiple Choice | | Fill-in-the-Blank | | True/False | | Desc. (Closed) | |
|---|---|---|---|---|---|---|---|---|
| | Simple | Difficult | Simple | Difficult | Simple | Difficult | Simple | Difficult |
| LLaVA-7B (Liu et al., 2023b) (Prompt) | 0.786 | 0.684 | 0.813 | 0.647 | 0.908 | 0.784 | 0.892 | 0.783 |
| LLaVA-13B (Liu et al., 2024b) (Prompt) | 0.829 | 0.705 | 0.883 | 0.673 | 0.920 | 0.888 | **1.000** | 0.913 |
| Qwen2-VL-7B (Wang et al., 2024) (Prompt) | 0.794 | 0.688 | 0.858 | 0.633 | 0.898 | 0.878 | 0.925 | 0.842 |
| Qwen2.5-VL-7B (Team, 2025) (Prompt) | 0.798 | 0.683 | 0.865 | 0.642 | 0.922 | 0.874 | 0.958 | 0.858 |
| Qwen2.5-VL-72B (team, 2025) (Prompt) | 0.875 | 0.714 | 0.882 | 0.677 | 0.920 | 0.878 | 0.992 | **0.950** |
| GPT-4o (Hurst et al., 2024) (Prompt) | 0.863 | 0.735 | 0.862 | 0.668 | 0.938 | 0.890 | 0.950 | 0.929 |
| Yo'LLaVA (Nguyen et al., 2024) (Raw) | 0.814 | 0.695 | 0.862 | 0.668 | 0.887 | 0.765 | 0.900 | 0.767 |
| Yo'LLaVA (Nguyen et al., 2024) (CoT) | 0.849 | 0.725 | 0.860 | 0.675 | 0.908 | 0.832 | 0.875 | 0.763 |
| LLaVA (Liu et al., 2023b) (Raw) | 0.865 | 0.762 | 0.863 | 0.753 | 0.893 | 0.840 | 0.850 | 0.796 |
| LLaVA (Liu et al., 2023b) (CoT) | 0.885 | 0.829 | 0.890 | 0.817 | 0.947 | 0.877 | 0.917 | 0.867 |
| **ReGraP-LLaVA (Ours)** | **0.942** | **0.892** | **0.940** | **0.858** | **0.967** | **0.916** | 0.975 | **0.950** |

Table 2: Comparison of ReGraP-LLaVA with prompt- and finetuning-based models on closed-ended QA tasks. The questions examining basic knowledge (e.g. features, recognition) are denoted as "Simple" and those requiring relational and multi-step reasoning are denoted as "Difficult".

## 6.1 CLOSE-ENDED QA

We feed raw personalized knowledge and CoT QA pairs in the ReGraP dataset separately to train both Yo'LLaVA and LLaVA, referred as Yo'LLaVA (Raw), Yo'LLaVA (CoT), LLaVA (Raw), and LLaVA (CoT), serving as finetuning-based models. For prompt-based models, we construct descriptions of personalized concepts in images using GPT-4o (prompts are detailed in Table 17), and manually verify that no direct answer leakage. These descriptions are then used to prompt the models.

Table 2 presents the accuracy results. ReGraP-LLaVA outperforms all baselines across most tasks, with the exception of simple descriptive QA, where LLaVA-13B (Prompt) achieves the highest accuracy of 1.000, followed by ours with a close third at 0.975. Moreover, LLaVA (CoT), trained on CoT QA pairs of our dataset, ranks second in 5 out of 8 tasks, further demonstrating the effectiveness of our data when training models. Numerically, our model achieves a large weighted improvement of 5.3% comparing to the best finetuning-based model, LLaVA (CoT), and 8.8% comparing to the best prompt-based model, GPT-4o (Prompt). In contrast, Yo'LLaVA, with its low computational overhead, performs well on simple tasks (e.g., basic recognition) but fails to capture complicated relational knowledge and reasoning processes due to limited learnable parameters.

To validate generalization beyond our own benchmark, we evaluate our model with Yo'LLaVA and MyVLM (Alaluf et al., 2024) using their datasets under identical settings. We prompt models with "Can you see <*concept name*> in this image?" for recognition tasks and "Caption this image in a short sentence." for captioning tasks. For captioning tasks, a response is considered correct if it includes the personalized concept and its meaning aligns with the content of the image. The evaluation includes both positive (concept-present) and negative (concept-absent) samples, where for negative samples, the correct answer is expected to be a denial (e.g., "no"). Table 3 shows the results. ReGraP-LLaVA showcases clear advantages in both positive tasks and comparable accuracy in negative tasks. The slightly lower accuracy in the negative recognition task may caused by few negative or counterfactual examples in training data, which makes our model more challenging to say "no" confidently.

## 6.2 OPEN-ENDED QA

We conduct experiments on open-ended descriptive tasks with 2 evaluation metrics. For each question, we construct 3 to 5 key points and assess whether models' responses cover them by both GPT-4o and human judges, and the resulting score, denoted as **Point** in Table 4, is computed by the number of matched points divided by the number of total points. Then, we employ GPT-4o with personalized knowledge and images to generate reference answers and subsequently perform as an evaluator to score model outputs based on the reference (prompts are detailed in Table 18 and Table 19).

Table 4 shows the results. Our model achieves the best performance in 3 out of 4 open-ended tasks and attains the highest scores in the Point metric for both tasks. For GPT-score on the full description task, ReGraP-LLaVA ranks a close second behind LLaVA-13B (Prompt). Notably, the Point metric offers the most substantive and quantitative assessment of the performance, while the GPT-Score serve as a reference, which reflects the alignment with GPT's styles and preferences rather than an absolute measure of answer quality. In this scenario, prompt-based methods receive GPT-generated descriptions as direct inputs, which influences their output style and may contribute to higher scores.

| Model | Recognition Accuracy | |
|---|---|---|
| | Positive | Negative |
| Yo'LLaVA | 0.925 | **0.857** |
| MyVLM | 0.905 | 0.823 |
| ReGraP-LLaVA | **1.000** | 0.850 |

| Model | Captioning Accuracy | |
|---|---|---|
| | Positive | Negative |
| Yo'LLaVA | 0.905 | 0.966 |
| MyVLM | 0.895 | 0.946 |
| ReGraP-LLaVA | **0.965** | **0.973** |

Table 3: Accuracy comparison on recognition and captioning tasks.

| Model | Desc. (Open) | | Full Desc. | |
|---|---|---|---|---|
| | Point | GPT-Score | Point | GPT-Score |
| LLaVA-7B (Prompt) | 0.729 | 9.06 | 0.951 | 9.48 |
| LLaVA-13B (Prompt) | 0.779 | 9.24 | 0.953 | **9.66** |
| Qwen2-VL-7B (Prompt) | 0.781 | 9.15 | 0.928 | 9.09 |
| Qwen2.5-VL-7B (Prompt) | 0.786 | 9.14 | 0.967 | 9.14 |
| Qwen2.5-VL-72B (Prompt) | 0.847 | 9.32 | 0.970 | 9.34 |
| Yo'LLaVA (Raw) | 0.661 | 8.46 | 0.916 | 9.06 |
| Yo'LLaVA (CoT) | 0.735 | 8.54 | 0.931 | 9.18 |
| LLaVA (Raw) | 0.705 | 8.34 | 0.896 | 9.12 |
| LLaVA (CoT) | 0.849 | 9.08 | 0.947 | 9.21 |
| **ReGraP-LLaVA (Ours)** | **0.878** | **9.36** | **0.978** | 9.49 |

Table 4: Performance on open-ended descriptive questions (Desc. (Open)) and detailed image description (Full Desc.).

## 6.3 ABLATION STUDY

In this section, we examine how the two proposed graph-prompting methods in section 4 influence model performance. We assess hard- and soft-prompt methods individually and also in combination to learn both respective and joint effects, and select the close-ended QA as the metric. Table 5 showcases the result. Overall, the single hard-prompt method achieves the highest accuracy, yielding 16 more correct answers than the combination method and 28 more than the single soft-prompt method out of 8160 questions. The slight accuracy difference across these methods (less than 0.4%) demonstrates the feasibility of all methods. Due to the marginal advantage of the hard-prompt method, we adopt it as the main method in other experiments. Notably, LLaVA (CoT) serves as an ablated variant of ReGraP-LLaVA without the graph-prompting module, and the performance gains over it in Sections 6.1 and 6.2 prove the effectiveness of graph-promptings. Besides, we conduct additional ablation studies on the number of objects and length of CoT QA pairs, which are detailed in section A.

| Model | Multiple Choice | | Fill-in-the-Blank | | True/False | | Desc. (Closed) | |
|---|---|---|---|---|---|---|---|---|
| | Simple | Difficult | Simple | Difficult | Simple | Difficult | Simple | Difficult |
| ReGraP-LLaVA (Hard) | **0.942** | 0.892 | 0.940 | 0.858 | **0.967** | 0.916 | 0.975 | 0.950 |
| ReGraP-LLaVA (Soft) | 0.938 | 0.893 | 0.938 | 0.852 | **0.967** | 0.910 | 0.942 | 0.929 |
| ReGraP-LLaVA (Combination) | 0.929 | **0.898** | 0.943 | 0.860 | 0.960 | 0.917 | 0.983 | 0.950 |

Table 5: Ablation Study. Comparison of the performance of different graph prompting methods of ReGraP-LLaVA on closed-ended QA tasks.

## 6.4 CASE STUDY

In Table 6, we showcase qualitative examples of model outputs across different query types. We compare our method with representative baselines: Yo'LLaVA (CoT), LLaVA (CoT), and LLaVA-13B (Prompt). All approaches succeed in basic recognition tasks. However, Yo'LLaVA sometimes fails to leverage the relations between personalized concepts and provides incorrect or overly generic responses in personalized scenarios. LLaVA (CoT) provides answers with high accuracy, while the

| The input image and attention visualizations of ReGraP-LLaVA | | | | |
|---|---|---|---|---|
| 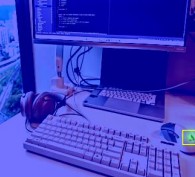 | 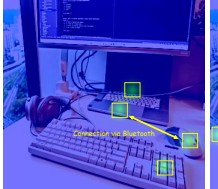 | 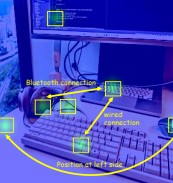 | | From left to right, images are the input and attention maps of Q1–3, respectively. Attention masks show that ReGraP-LLaVA attends to task-related objects, aligning with the semantic content of its responses. |

**Question 1**: What is the brand of my mouse? (Query requiring knowledge of features of an object)

| **Yo'LLaVA (CoT)**: The brand of the mouse in the image is Logitech. I can identify it through the following steps: 1. **Logo Recognition**: The mouse has a distinctive logo on the side... This logo is universally recognized as the brand logo for Logitech... | **LLaVA (CoT)**: The mouse is an Asus ROG mouse. I know this because Asus ROG...The mouse has a distinctive design with a premium finish... Since it is an Asus ROG mouse... | **LLaVA-13B (Prompt)**: The mouse in the image is from Logitech, as indicated by the logo on the mouse. The logo consists of the word "Logitech" in lowercase letters, with the "o" in "Logitech". This is a common logo for Logitech products,... | **ReGraP-LLaVA**: The mouse is an Asus ROG mouse. The mouse has a distinctive logo which looks like an eye. This logo is universally recognized as the brand logo for Asus ROG. |
|---|---|---|---|

**Question 2**: How can the owner improve the gaming experience, and why? (Query requiring relational reasoning)

| **Yo'LLaVA (CoT)**: To improve the gaming experience, the owner could consider the following: 1. Ergonomics:...2. Monitor:...a higher refresh rate...3. Headphones:...Wireless headsets...4. Keyboard and Mouse:...consider mechanical keyboards with programmable keys and a mouse with adjustable sensitivity and DPI settings... | **LLaVA (CoT)**: The owner has...Using a wired connection reduces latency, ensuring smooth gaming without lag. This improves the gaming experience by providing high-quality, real-time audio output without concerns about wireless latency. | **LLaVA-13B (Prompt)**: To improve the gaming experience...: 1. Ergonomics: Ensuring that the keyboard and mouse are at a comfortable angle and...2. Cable Management:...3. Monitor Position: Adjusting the monitor to the optimal viewing angle and distance...4. Headphones:....5. Cooling:... 6. Gaming Accessories:...7. Software Settings:...8. Comfortable Seating:... | **ReGraP-LLaVA**: ...The owner should switch the Asus ROG mouse from a Bluetooth connection to a wired connection. Currently, the mouse is connected wirelessly via Bluetooth to the Asus ROG laptop, which introduces slight latency. For gaming ... a wired connection would offer more stability and lower latency, improving ... The Filco keyboard is in a 45-degree position, which allows for more comfortable wrist positioning during prolonged use... |
|---|---|---|---|

**Question 3**: How should I arrange the desktop? (Query requiring overall understanding)

| **Yo'LLaVA (CoT)**: In the image, you have a desktop setup ... To arrange the desktop for a left-handed person, you would want to ensure that the keyboard and mouse are placed on the right side of the desk, as... The monitor should be positioned to the left of the keyboard and mouse, so that ... | **LLaVA (CoT)**: ...To optimize the workspace for a left-handed person, start by positioning the keyboard and mouse closer to the center. Then, arrange the monitor directly in front of you, and finally, place the headphones on the left side of the desk... | **LLaVA-13B (Prompt)**: ...you have a well-organized workspace with a desktop computer setup. ... which is typical for right-handed users. To adjust the keyboard angle, you can use a keyboard tray or a riser to...For the mouse, you can adjust its position to the right of the keyboard to...The headphones are also positioned in a way that they are easily accessible... | **ReGraP-LLaVA**: ... arranged for a left-handed user. The keyboard is wired and connected to the laptop and can be placed on the right side of the desk at a 45-degree angle, the mouse... the left side of the desk, and the headphones are connected to the laptop via Bluetooth and positioned at the left side of the desk. The monitor is used for...and placed at the left side... |
|---|---|---|---|

Table 6: Qualitative comparison with other methods. We use green to denote correct and expected contents, blue to denote irrelevant or unexpected contents, and red to denote wrong contents. Attention visualizations of ReGraP-LLaVA further demonstrate that the model grounds its reasoning on learned personalized knowledge, focusing on regions related to task-specific question answering.

reasoning process is occasionally unexpected or even incorrect. It also includes irrelevant information in its responses. LLaVA-13B (Prompt) exhibits similar drawbacks with LLaVA (CoT), and is impacted by low-quality prompts. In contrast, ReGraP-LLaVA consistently provides correct and contextual responses, and the detailed responses illustrate that it utilizes relational knowledge. The visualization of attentions of ReGraP-LLaVA further demonstrates that it recognizes and focuses on task-related objects (regions). More qualitative examples of ReGraP-LLaVA are detailed in section H.

## 7 CONCLUSION

In this work, we leverage knowledge graphs and CoT QA pairs to enhance the reasoning capabilities of MLLMs for personalization. We introduce the ReGraP dataset and a novel MLLM, ReGraP-LLaVA, which is trained on images, CoT QA pairs and soft and/or hard prompts of knowledge graphs. We investigate the feasibility of both soft and hard prompts in training, and establish the ReGraP benchmark to evaluate models' relational reasoning and knowledge connection capability on personalized knowledge. Experimental results show that ReGraP-LLaVA effectively learns personalized knowledge and utilize it for accurate and contextual answers, which demonstrates the effectiveness of both our dataset and methods. Future works can explore more effective methods to align knowledge graphs with MLLMs at low cost while preserving the capability of personalization.

## REPRODUCIBILITY STATEMENT

We ensure the reproducibility of our research. We publicly release the dataset and code, and parts of the data have been made public in the anonymous link in the abstract. The prompts in data generation pipeline and evaluation are present in the section E in Appendix. We also present numerous qualitative examples in the section F and section G in Appendix.

## ETHICS STATEMENT

The main artifact of this work is the human evaluation on models' responses and quality of the data, which are detailed in section B and section C in Appendix. We engage three experienced NLP researchers as annotators, and approximately 50 human hours are devoted. All annotators are properly briefed, and compensated at $20 per hour, which is above the local average. Both the dataset and code are made public under the MIT license to encourage further researches.

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

## A   ADDITIONAL ABLATION STUDIES

**Number of personalized objects in a set.** We investigate how the number of personalized objects in a set influences model performance. To this end, we construct 2-, 3-, and 4-object sets by reducing the number of personalized concepts from multi-object sets, and remove the corresponding knowledge from both the KGs and the CoT QA pairs. We then conduct experiments on close-ended questions, selecting those pertaining to the remaining objects. Figure 4 showcases the results. Overall, the accuracy remains consistent across different object settings, which suggests that the number of objects has a limited impact on model performance. The key reason lies in our method: regardless of the number of objects, we construct KGs and CoT QA pairs by fully utilizing attributes and relations of each object. This ensures the model receives complete knowledge.

**Length of CoT answers in CoT QA pairs.** We investigate how the length of answers in CoT QA pairs influences model performance. To this end, we refine the long answers by reducing the reasoning steps. We then conduct experiments on close-ended questions. Figure 5 showcases the results. Overall, the accuracy on simple tasks remains largely unaffected by the reduced answer length. However, for difficult tasks that require multi-step reasoning and relational inference, performance improves as the length of the answers increases. This indicates that longer CoT answers play a critical role in supporting the model's reasoning capability for challenging queries, while providing fewer benefits for basic recognition and knowledge acquisition.

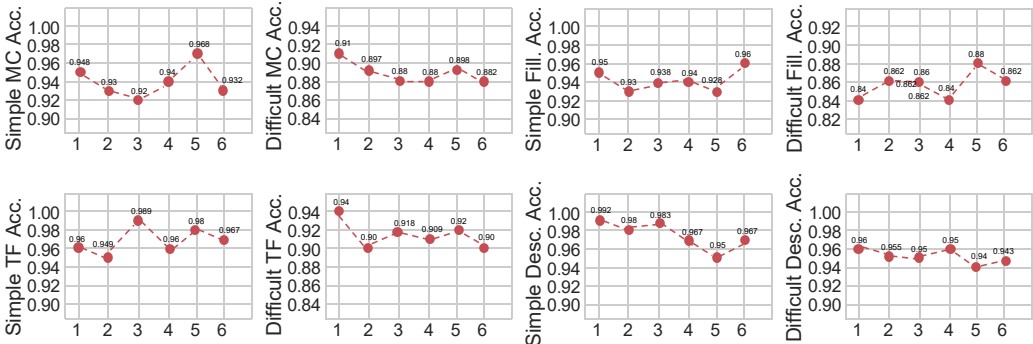

Figure 4: The ablation study of the number of personalized objects.

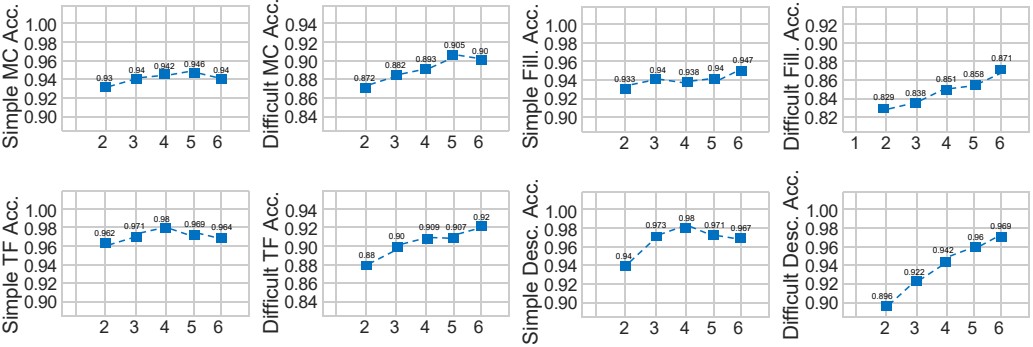

Figure 5: The ablation study of the length of answers in CoT QA pairs.

## B   HUMAN EVALUATION

We investigate how ReGraP-LLaVA aligns with human's preference. Inspired by LIMA (Zhao et al., 2024), we adopt two human evaluation metrics: (1) Given a set of personalized knowledge, a question and two responses from ReGraP-LLaVA and a baseline respectively, we ask human annotators to judge if "response 1 is better" (ReGraP-LLaVA wins), "cannot tell difference" (Tie) or "response 2 is better" (Baseline wins). (2) Given a set of personalized knowledge and two MLLMs, we ask human annotators to ask questions to the models respectively, and tell if "model 1 is better" (ReGraP-LLaVA

wins), "cannot tell difference" (Tie) or "model 2 is better" (Baseline wins). Figure 6 (a) shows the results of the first metric and Figure 6 (b) shows the results of the second metric. The results demonstrate that our model gives responses aligning with human's preference better.

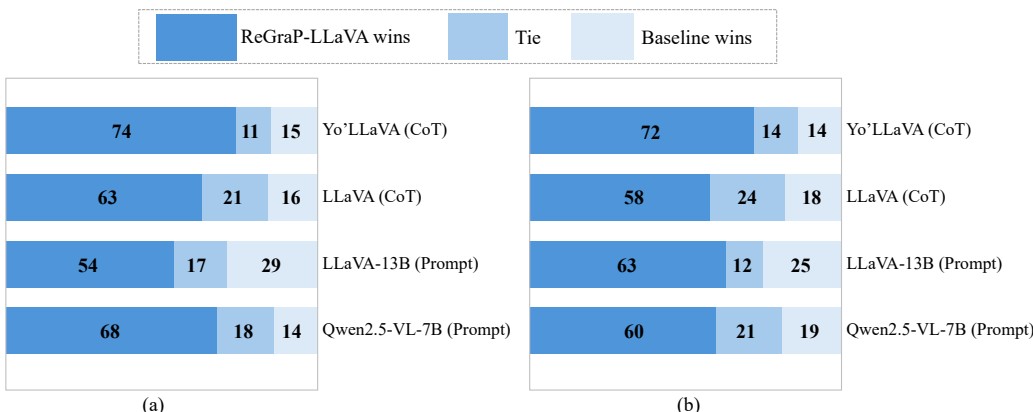

Figure 6: The human evaluation results. The numbers are the counts of each case.

## C  EVALUATION ON THE QUALITY OF CoT QA PAIRS

In this section, we evaluate the quality of the CoT QA pairs using both human annotators and LLMs. The evaluation is conducted across four questions: (1) *Does the answer address the question*, (2) *"Is the reasoning correct in logic?"*, (3) *Does the answer demonstrate step-by-step reasoning that leads to the correct conclusion*, and (4) *Is the overall quality of the response satisfactory*, which are denoted as "accuracy", "logic", "reason" and "quality" separately. For the evaluation, we randomly sample 200 CoT QA pairs, each of which is independently assessed across the above criteria. Evaluators are instructed to provide binary judgments (*Yes* or *No*) for each question. The LLMs employed for evaluation include GPT-4o-2024-11-20, Qwen-Max-2025-01-25, Grok-3, and Deepseek-R1. Table 7 presents the evaluation results. Notably, all LLMs except GPT-4o consistently response "Yes" across all questions, reflecting a strong alignment with their preferences. For the "reason" criterion, GPT-4o occasionally provides a "No" judgment, and it explains that these answers reveal the conclusion at the start rather than arriving at it after step-by-step reasoning. The same reason also appears in other evaluations and leads to an assignment of "No". Nevertheless, both GPT-4o and humans acknowledge the overall quality of these QA pairs, suggesting that this judgment pertains to the formats rather than a deficiency in its content or correctness. Table 8 shows the qualitative example of this case.

| Evaluator | Accuracy (Yes%) | Logic (Yes%) | Reason (Yes%) | Quality (Yes%) |
|---|---|---|---|---|
| GPT-4o-2024-11-20 | 100 | 99 | 88 | 100 |
| Qwen-Max-2025-01-25 | 100 | 100 | 100 | 100 |
| Grok-3 | 100 | 100 | 100 | 100 |
| Deepseek-R1 | 100 | 100 | 100 | 100 |
| Human | 99 | 100 | 94 | 98 |

Table 7: Evaluation of CoT QA pairs by different LLMs and humans across four binary-questions.

| Question: | "Is <anime_calendar> portable? |
|---|---|
| Answer: | Yes, the small and lightweight nature of <anime_calendar> makes it portable. The compact size means it can easily be moved around and placed in different areas, whether it is on a desk, shelf, or countertop. This makes it ideal for people who like to reposition or take it with them when needed. Thus, <anime_calendar> is portable." |

Table 8: Qualitative example of a negative assignment case.

# D  ADDITIONAL RELATED WORK

**Chain-of-Thought Reasoning in Multimodal tasks.** MM-CoT (Zhang et al., 2024) studies the CoT reasoning in different modalities, and proposes a two-stage framework to perform Multimodal-CoT. The model first generates rationales and subsequently inference final answers. Experimental results demonstrate the effectiveness of this multi-step reasoning process in vision-related tasks.

CCoT (Mitra et al., 2024) proposes a zero-shot CoT approach that utilizes scene graphs to extract compositional knowledge out of an MLLM. The model is first instructed to construct a scene graph containing the objects, their attributes, and the relations in an image. Thereafter, the scene graph is converted into text, and included in a follow-up prompt jointly with the original question to produce the final answer. Experimental results demonstrate the performance gain brought by the graph-prompting.

Visual CoT (Shao et al., 2024) introduces a visual CoT dataset and a multi-turn inference pipeline for MLLMs. The model first attends to highlighted regions and generates an interpretable thought, then progressively improves its answer. Experimental results demonstrate the framework's effectiveness on the improvement of the model's visual understanding ability.

**Integrating Knowledge Graphs in MLLMs.** KAM-CoT (Mondal et al., 2024) proposes the Knowledge Augmented Multimodal CoT approach, KAM-CoT, which injects knowledge graphs into the reasoning process of models. Experimental results demonstrate that the incorporation of KG in the two-stage training process helps reduce hallucination and provide accurate answers.

GraphVis (Deng et al., 2024) proposes a method to convert KGs into visual promptings in order to instruct the model to learn knowledge in an image thoroughly. This method enhances LLMs' ability to learn and reason over KG data to enhance the textual QA responding. Experimental results demonstrate notable accuracy gains on knowledge-intensive visual QA tasks.

# E  HYPERPARAMETERS AND PROMPTS

In this section, we present the hyperparameters to train the model. Table 9 showcases the details. We also present the prompts in knowledge graph construction (Table 13, Table 14, Table 15, Table 16) and in performance evaluation (Table 17, Table 18, Table 19). Specifically, the GPT version in this work is GPT-4o-2024-11-20. All responses are manually reviewed, and necessary adjustments (e.g., formatting in data generation stage, few-shot correction for description generation) are applied to ensure data quality and robustness in both generation and performance assessment stages.

| Category | Hyper-parameter | Value |
|---|---|---|
| LoRA | Rank $r$ | 8 |
| | Lora_alpha $\alpha$ | 16 |
| | Target modules | {"q_proj", "v_proj"} |
| | Dropout | 0.1 |
| | Bias | default |
| | Task type | CAUSAL_LM |
| AdamW Optimizer | Learning rate | $1 \times 10^{-5}$ |
| | Betas | (0.9, 0.999) |
| | Weight decay | 0.1 |
| | Epsilon | $1 \times 10^{-8}$ |
| | Fused | True |
| LR Scheduler (optional) | Strategy | CosineAnnealingLR |
| Mixed precision | – – | GradScaler |
| Hardware | GPU | $1 \times$ NVIDIA A6000 |

Table 9: Hyperparameters setup.

## F TRAINING DATA EXAMPLES

In this section, we present representative examples of the training data, which include images, the procedures for constructing KGs, and corresponding examples of CoT QA pairs. Table 20, Table 21, and Table 22 illustrate examples from single-object sets, where we focus on a single object and leverage its attributes and components to construct the knowledge graph. Table 23, Table 24, and Table 25 illustrate examples from multi-object sets, where multiple objects or characters, along with their attributes and relations, are integrated to build the knowledge graph.

## G BENCHMARK EXAMPLES

In this section, we present representative examples of our benchmark, which include Multiple-Choice, Fill-in-the-blank, True/False, and Descriptive questions. Table 10 presents a comparison between our benchmark and the Yo'LLaVA benchmark. We establish questions that not only assess basic recognition and captioning abilities, but also emphasize understanding of relations among multiple personalized concepts and utilize the learned knowledge in answering. Table 26, Table 27, Table 28, Table 29, Table 30, and Table 31 showcase the different types of answers and reference answers in detail. The benchmark encompasses both simple questions focused on attribute learning or recognition and more complex questions that require reasoning over relations and detailed image understanding.

| Aspect | Yo'LLaVA | ReGraP (Ours) |
|---|---|---|
| **Question Types (Closed-Ended)** | | |
| Text-only | ✓ | ✓ |
| Visual | ✓ | ✓ |
| Binary Choice | ✓ | ✓ |
| Multiple-Choice (opt. > 2) | ✗ | ✓ |
| Fill-in-the-Blank | ✗ | ✓ |
| Recognition | ✓ | ✓ |
| Descriptive | ✗ | ✓ |
| **Open-Ended Evaluation** | | |
| Captioning | ✓ | ✓ |
| Open Descriptive QA | ✗ | ✓ |
| Overall Description | ✗ | ✓ |
| **Evaluation Features** | | |
| Difficulty Diversity | ✗ | ✓ |
| Multi-object QA | ✗ | ✓ |
| Relational Reasoning | ✗ | ✓ |
| Reference Answers | ✗ | ✓ |

Table 10: Comparison of benchmark designs between Yo'LLaVA and ReGraP.

## H QUALITATIVE RESULTS

In this section, we present qualitative examples that showcase ReGraP-LLaVA's question-answering capabilities. Table 32, Table 33, Table 34, Table 35, Table 36, and Table 37 provide the questions and model's answers in detail. The responses demonstrate that ReGraP-LLaVA effectively learns personalized knowledge and accurately utilizes it to handle user queries across different types of questions. The model is able to provide detailed analyses of images and leverage the corresponding learned knowledge to generate contextually appropriate and comprehensive answers.

## I DISCUSSION ON ERRORS AND DEVIATIONS

Due to the inherent stochasticity in both the training and inference processes of MLLMs, we conduct a robustness analysis to evaluate the stability and reliability of our results. The main concern is whether repeated training runs with the same data and experimental settings introduce variability in

the model's outputs. We select five personalized knowledge sets and train five models independently from scratch for each set. Subsequently, we prompt each model with the same image and a query corresponding to its associated personalized concept: "Give a short description on *<concept name>* in this image.". We assess whether generated outputs exhibit semantic consistency across models trained on the same training set. Encouragingly, the outputs remain semantically consistent across models, confirming the robustness of our training pipeline. Table 11 presents qualitative examples.

Another main concern is the reproducibility of the model's outputs. For each model trained on a distinct personalized knowledge set, we prompt it 10 times with the same image and a query corresponding to its associated personalized concept: "Give a short description on *<concept name>* in this image.". We examine whether the generated responses remain semantically consistent across runs. Encouragingly, all outputs are consistent in content, with the vast majority being totally identical, which demonstrates the strong robustness in the inference process.

| | |
|---|---|
| Input image: | 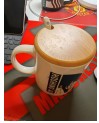 |
| Response 1: | The image features a close-up of a cup with a spoon, indicating that it is a beverage item. The cup is filled with a light-colored liquid, likely tea, suggested by its calming tone. The wooden lid is placed neatly on top, adding to the cozy atmosphere. The anime character print is well-defined, with vibrant colors that make the design attractive. Thus, the image depicts a notably designed cup with an anime character print, intended for beverage consumption and collection. |
| Response 2: | The image features a close-up of a cup with a spoon resting on a wooden lid, suggesting that this is a beverage cup designed for stirring. The anime character next to the cup adds a personalized touch, making it a unique gift or item of interest. The combination of wooden material and illustration suggests that it is not just a functional cup but also a collectible item. Thus, the image depicts a notably designed cup with both functional and collectible elements. |
| Response 3: | The image features a close-up of a cup with a spoon, indicating that it is a beverage item. The anime character next to the cup adds a personalized touch, suggesting that it is used by a fan. The combination of these elements confirms that this is a notably designed cup with an anime character print, intended for hot beverages by a fan of the genre. Thus, the image describes <anime_cup> as a notably designed cup with an anime character print. |
| Response 4: | The image features a close-up of a cup with a spoon and a wooden lid, suggesting that it is a beverage item. The wooden lid is in place, adding to the cozy atmosphere. The anime character print adds a personalized and artistic touch to the item. Thus, the image depicts a notably designed cup with an anime character print. |
| Response 5: | The image features a close-up of a cup with a wooden lid and a spoon, suggesting that it is a beverage item. The wooden lid is in place, adding to the cozy appeal. The anime character is depicted in vibrant colors, smiling and holding the cup, enhancing its appeal. Thus, the image suggests that this is a notably flavored cup with an anime-themed print, designed for beverages and collection. |

Table 11: Qualitative example of deviations. The responses remain consistent.

## J LIMITATIONS.

ReGraP-LLaVA is not without limitations. First, due to the inherent hallucination in language models, it may occasionally generate fabricated information about the image or the queried entities (e.g., inventing features that do not exist). Second, ReGraP-LLaVA sometimes produces overly detailed responses to simple queries. While this may improve answers' accuracy, it can reduce computational efficiency and potentially diverge from human preferences, leading to responses that are less aligned with practical expectations. Table 12 showcases a qualitative example of an overly detailed answer.

| **Limitations** | |
| --- | --- |
| Input image: | 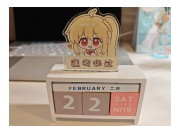 |
| Question: | Can you see <anime_calendar> is in this photo? |
| ReGraP-LLaVA: | From the image, it is possible to observe several elements: the anime character with yellow hair, the calendar date, and the anime figure itself. These details are clearly visible and contribute to the overall aesthetic of the item. The anime character's date is 'FEBRUARY 22' and the figure itself has a movable anime character. These elements are clearly seen and add to the visual appeal of the item. Thus, one can see the anime character, the date, and the movable feature. The image allows for a full view of these elements. The nature of the item means that one can see the character, date, and movable feature clearly. Thus, what is visible in the image is consistent with what is available for use. Thus, the anime character, the date, and the movable feature are visible. Thus, **you can see the <anime_calendar>**, the anime character, the date, and the movable feature clearly. |

Table 12: Qualitative example of limitation. The response from ReGraP-LLaVA is very detailed and shows that the model learns the personalized knowledge well. However, user may just need a short answer (e.g., "yes" or "no"), instead of a long paragraph.

## K USAGE OF LLMS

We hereby disclose the usage of large language models (LLMs) in this paper. As detailed in Section 3, Section 5, and Section 6.2, we employ LLMs both as data generators and as evaluators. The detailed prompts are presented in Section E. While LLMs play a significant role in the data generation pipeline, their role in evaluation is limited to serving as a supplementary component in open-ended tasks, which illustrate the generative capabilities of our models.

| Prompts of the Relation-Graph Builder |
|---|
| """
You are builder of a relation graph, you should be fair, read the images and input texts carefully and reflect all knowledge in them without any make-up information.
You focus on the relations of <personalized concepts> in the picture.
The relational knowledge of them is:
[Relational Knowledge]\n""" + <textual description> + "\n" + """
Your output should be in Json format, you should firstly generate a list containing information of all nodes, and then a list of edges. Each edge should strictly follow this format: "from": <entity 1>, "to": <entity 2>, "relation": <relation 1>
Here is an example of output format:
{
"nodes": [
{"id": "Bocchi", "type": "Person", "school": "A", "grade": "1", "role": "Band Member"},
{"id": "Ryo", "type": "Person", "school": "B", "grade": "1", "role": "Band Member"},
{"id": "Nijika", "type": "Person", "school": "B", "grade": "2", "role": "Leader"},
{"id": "Kita", "type": "Person", "school": "A", "grade": "2", "role": "Band Member"},
{"id": "Kessoku Band", "type": "Band"}
],
"edges": [
{"from": "Bocchi", "to": "Kessoku Band", "relation": "is guitarist of "},
{"from": "Kita", "to": "Bocchi", "relation": "is potential couple of"},
{"from": "Ryo", "to": "Kita", "relation": "is a senior of"},
]
}
Your output must be strictly structured in the above JSON format.
Your answer is:
""" |

Table 13: Prompts of the Relation-Graph Builder

| Prompts of the Relation-Graph Builder |
|---|
| """
You are KG Enricher which enrich the input knowledge graph while keeping its format, you should be fair, read the images and input texts carefully and reflect all knowledge in them without any make-up information.
You focus on the <personalized concepts> and their attributes. Your job is:
(1)Add nodes representing attributes and potential new concepts to the graph.
(2)Explore and add new edges representing relations between the nodes.
The input knowledge graph that to be enriched is:
[Input Knowledge Graph]\n""" + <KG> + "\n" + """
The attribute-based knowledge is:
[Knowledge]\n""" + <textual description> + "\n" + """
Your output must be strictly structured in the JSON format of the input file.
Your answer is:
""" |

Table 14: Prompts of the KG Enricher

| Prompts of the CoT Question Generation |
|---|
| """
Given the following reasoning steps and personalized knowledge, follow the example, you should generate a question that takes the reasoning steps as the thinking process to reach the answer. The question should not be too simple and should require relational reasoning.
The reasoning steps are:
[Reason steps]\n""" + <R> + "\n" + """
The personalized knowledge is:
[Personalized knowledge]\n""" + <personalized knowledge> + "\n" + """
The example is:
[Example]\n""" + Reasoning steps:... Question:... + "\n" + """
The question is:
""" |

Table 15: Prompts of the CoT Question Generation

| Prompts of the CoT Answer Generation |
|---|
| """
Given the following reasoning steps and the question, you should refine the reasoning steps and give a comprehensive, styep-by-step, and full CoT answer, which reflects all information to reach the answer of the question.
The reasoning steps are:
[Reason steps]\n""" + <R> + "\n" + """
The question is:
[Question]\n""" + <question> + "\n" + """
The refined answer is:
""" |

Table 16: Prompts of the CoT Answer Generation

| Prompts of Personal knowledge description |
|---|
| """
You are a personalized knowledge descriptor, you job is to give a short description for the overall input image and each personalized entity in the image, based on both visual input and additional knowledge.
The personalized entities are:
[Personalized entities]\n""" + <textual description> + "\n" + """
The Additional Knowledge is:
[Additional Knowledge]\n""" + <textual description> + "\n" + """
Your output must be strictly structured in the following JSON format:
{
"Image Description": "<Text>",
"<Entity 1> Description": "<Text>"
"<Entity 2> Description": "<Text>"
...
}
""" |

Table 17: Prompts of Personal knowledge description

| Prompts of Reference Answer Generation |
|---|
| """
You are a Reference Answer Generator, your job is to generate a reference answer for the input question. You should carefully consider the input image and the related knowledge, and DO NOT make up any information. Your answer should show step-by-step thinking process that eventually reach the answer.
The Question is:
[Question]\n""" + \<question> + "\n" + """
The related knowledge is:
[Related Knowledge]\n""" + \<textual description> + "\n" + """
Please think step by step, and use a concise tone for each step of thinking.
Your answer is:
""" |

Table 18: Prompts of Reference Answer Generation

| Prompts of Open-ended Assessment |
|---|
| """
You are a strict and objective evaluator. Your task is to assess the quality of the answer for the input question. Your evaluation should be fair, professional, and reflect an expert judgment of the answer's quality.
You should carefully consider the input image and the related knowledge, and also refer to the reference answer. Your assessment should strictly follow the assessment criteria.
The Question is:
[Question]\n""" + \<question> + "\n" + """
The related knowledge is:
[Related Knowledge]\n""" + \<textual description> + "\n" + """
The reference answer is:
[Reference]\n""" + \<reference> + "\n" + """
The model's answer is:
[Answer]\n""" + \<answer> + "\n" + """
Assessment Criteria:
Completeness: Does it shows a complete reasoning process to reach the answer?
Correctness: Does it correctly solve the question? Does it leverage useful information both from the picture and related knowledge?
Politeness: Does the answer keeps in a polite tone?
Weakness: Does it makes up any information or talk about irrelevant information? If so, the score can be reduced by the degree.
Overall Score: Assign a score from 1 to 10, 10 is the best, and a correct answer should not be assigned with a score less than 7 unless it encounters significant factual error. Considering all of the above factors.
The evaluation and your output must be strictly structured in the following JSON format:
{
"Explanation": "\<Explain the rationale of your score.>",
"Score": \<An integer score from 1 to 10.>
}
""" |

Table 19: Prompts of Open-ended Assessment

**<anime_cup>**

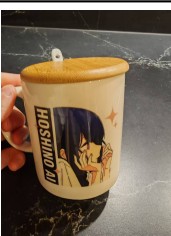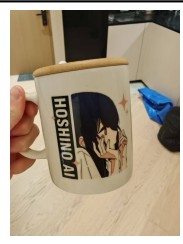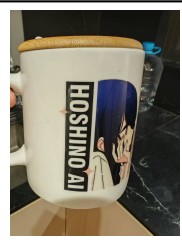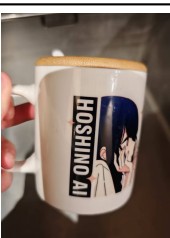

Objects: cup is the single object, its lid, character on the cup, spoon and corresponding attributes of these items help construct the knowledge graph.

**Q:** What is the object of `<anime_cup>`?
**A:** By closely examining the image, `<anime_cup>` is identified as a ceramic cup featuring an anime-themed design. The presence of a handle, a wooden lid, and a spoon suggests that it is designed for everyday beverage consumption. The sturdy build and artistic elements indicate that it is not just a functional item but also a collectible or decorative piece. Thus, `<anime_cup>` is a ceramic cup with an anime character print, designed for both use and display.
**Q:** What is the primary color of `<anime_cup>`?
**A:** `<anime_cup>` is primarily white, which serves as the base color for the entire ceramic surface. The white background enhances the clarity and vibrancy of the printed anime illustration, making it visually appealing. The contrast between the dark-haired character and the white ceramic surface highlights the artistic design. Thus, `<anime_cup>` has a predominantly white ceramic body.
**Q:** What is the name of the character printed on `<anime_cup>`?
**A:** The anime character featured on `<anime_cup>` is Hoshino Ai, as indicated by the text next to the illustration. The character is well-recognized within anime culture, adding to the appeal of the cup for fans and collectors. Thus, `<anime_cup>` features the character Hoshino Ai.
**Q:** What is the material of the cover on `<anime_cup>`?
**A:** The cover of `<anime_cup>` is made of natural wood, as indicated by its visible grain texture and light brown color. Wooden lids are commonly used in high-quality mugs to provide insulation and an eco-friendly alternative to plastic covers. This material choice also adds a rustic and elegant aesthetic to the cup. Thus, `<anime_cup>` has a wooden lid made of natural wood.
**Q:** Does `<anime_cup>` have a cover?
**A:** Yes, `<anime_cup>` includes a wooden cover that fits securely on top. This lid serves multiple purposes, such as keeping the beverage warm, preventing dust or external particles from contaminating the drink, and adding an aesthetic contrast to the white ceramic body. The presence of a notch in the lid also suggests it is designed to accommodate a spoon. Thus, `<anime_cup>` has a functional wooden cover.
**Q:** Does `<anime_cup>` come with a spoon?
**A:** Yes, `<anime_cup>` includes a spoon that appears to be placed inside. The presence of a notch in the wooden lid suggests that the spoon is intended to be used alongside the cup, making it ideal for beverages that require stirring, such as tea or coffee. This addition enhances its usability. Thus, `<anime_cup>` includes a spoon designed for stirring beverages.
**Q:** Is `<anime_cup>` suitable for hot beverages?
**A:** Yes, `<anime_cup>` is well-suited for hot beverages. The ceramic material is known for its heat-retaining properties, which help keep drinks warm for an extended period. The handle ensures safe handling by preventing direct contact with the heated surface. Additionally, the wooden lid provides insulation, further preserving the drink's temperature. Thus, `<anime_cup>` is designed for hot beverages such as tea and coffee.
......

Table 20: The training example of <anime_cup>

**<anime_calendar>**

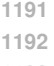 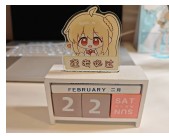 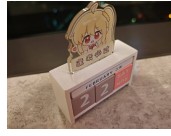 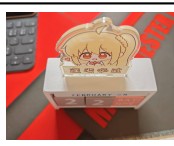 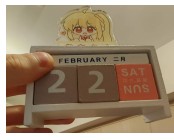 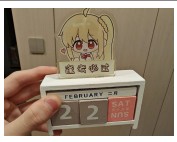

Objects: The object is a wooden calendar featuring an anime character (Nijika) with yellow hair on top, movable date blocks, and bilingual (English and Chinese) words. It has a cuboid structure and serves both functional (read the dates) and decorative (anime style) purposes.

**Q:** What is the object of `<anime_calendar>`?
**A:** By observing the image, the object is clearly a calendar. It consists of a rectangular base with movable date blocks and a character figure on top, functioning as a traditional date-keeping device. Thus, `<anime_calendar>` is a calendar.

**Q:** What color is `<anime_calendar>`?
**A:** The calendar's primary colors are white, yellow, and pink. The base is white, the movable blocks are pink, and the anime character has yellow hair, creating a colorful and visually appealing design. Thus, `<anime_calendar>` is white, yellow, and pink.

**Q:** What is the material of `<anime_calendar>`?
**A:** The material of `<anime_calendar>` appears to be wood, based on its solid and smooth texture. This ensures durability and gives a natural aesthetic to the calendar. Thus, `<anime_calendar>` is made of wood.

**Q:** What is the pattern of `<anime_calendar>`?
**A:** The pattern features an anime character with yellow hair on the top section, along with date blocks below that display numbers and characters. This gives it a unique and decorative appearance suitable for anime fans. Thus, `<anime_calendar>` has a pattern of an anime character.

**Q:** What languages are used on `<anime_calendar>`?
**A:** The calendar uses English ("FEBRUARY"), Chinese (day labels), and numerical figures for dates. This bilingual design makes it accessible for both English and Chinese readers. Thus, `<anime_calendar>` uses English, Chinese, and figures.

**Q:** What is the design style of `<anime_calendar>`?
**A:** The design blends a functional cuboid shape with playful anime-themed decoration. It combines simplicity in structure with vibrant, aesthetic elements, appealing to users who appreciate both utility and style. Thus, `<anime_calendar>` follows a functional yet playful design style.

textbfQ: What is the stability of `<anime_calendar>`?
**A:** Observing its flat base and the wooden blocks resting securely within, it exhibits the traits of a stable structure. The solidity of its foundation prevents it from easily toppling over when the blocks are changed. This design ensures minimal disruption to the displayed date. Hence, `<anime_calendar>` is stable.

**Q:** Is `<anime_calendar>` portable?
**A:** Taking into account its small form factor and the practicality of wooden construction, it is simple to move or carry from one spot to another. The manageable dimensions mean it fits in various settings like desks, shelves, or counters. Despite being sturdy, it remains suitably compact. Consequently, `<anime_calendar>` is portable.

**Q:** What is the durability of `<anime_calendar>`?
**A:** Considering wood as the primary component, combined with careful assembly to handle frequent flipping of blocks, it is designed for long-term use. Frequent adjustments should not lead to rapid wear. This construction can endure daily contact without easily breaking. Thus, `<anime_calendar>` is durable.

Table 21: The training example of <anime_calendar>

**<ROG_mouse>**

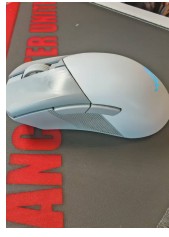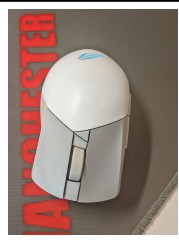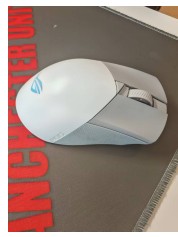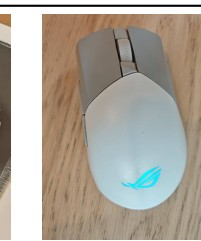

Objects: The object is a mouse primarily designed for gaming, and its brand is the ASUS Republic of Gamers (ROG). It features a compact, ergonomic design with customizable RGB lighting, wireless connectivity, and a smooth plastic body, optimized for precision and comfort.

**Q:** What object is `<ROG_mouse>`?
**A:** By observing the image, it is clear that the object shown is a computer mouse, designed for user interaction with a computer system. The shape, buttons, and scroll wheel confirm its purpose. Thus, `<ROG_mouse>` is a mouse.
**Q:** What color is `<ROG_mouse>`?
**A:** The mouse is primarily white with grey detailing. The body is white while the side grips and accents are grey, complemented by customizable lighting effects. Thus, `<ROG_mouse>` is white and grey.
**Q:** What is the material of `<ROG_mouse>`?
**A:** Based on its smooth and lightweight appearance, the mouse is made of plastic. This provides durability while keeping the device light enough for fast movements. Thus, `<ROG_mouse>` is made of plastic.
**Q:** Is `<ROG_mouse>` a wireless one or a wired one?
**A:** The absence of a visible cable and the overall design suggest that `<ROG_mouse>` is a wireless device, offering greater freedom of movement during gaming or work. Thus, `<ROG_mouse>` is a wireless mouse.
**Q:** What is the feature of `<ROG_mouse>`?
**A:** The mouse boasts customizable RGB lighting, wireless functionality, and an ergonomic form factor tailored for gaming. Despite signs of aging, it remains highly functional. Thus, `<ROG_mouse>` features RGB lighting, is wireless, and is in good condition.
**Q:** What is the texture of `<ROG_mouse>`?
**A:** The mouse surface appears smooth with a slight gloss on the white sections and a matte finish on the grey sections, providing comfort and grip during use. Thus, `<ROG_mouse>` has a smooth texture.
**Q:** What is the ergonomic design of `<ROG_mouse>`?
**A:** The mouse is ergonomically shaped for right-handed users, with contours that fit naturally into the hand, minimizing wrist strain and improving control during prolonged use. Thus, `<ROG_mouse>` features an ergonomic design for comfort.
**Q:** What is the brand of `<ROG_mouse>`?
**A:** The logo indicates that the mouse belongs to the ASUS Republic of Gamers (ROG) product line, known for high-performance gaming peripherals. Thus, `<ROG_mouse>` is a product of ASUS ROG.
**Q:** What type of lighting effects does `<ROG_mouse>` feature?
**A:** The mouse is equipped with RGB lighting effects, customizable via software, allowing users to personalize their setup with different colors and effects. Thus, `<ROG_mouse>` features customizable RGB lighting.
**Q:** What is the reaction time of `<ROG_mouse>`?
**A:** With a design aimed at gamers, the mouse achieves a very low reaction time, likely around 1 millisecond, ensuring near-instantaneous responses during gameplay. Thus, `<ROG_mouse>` has a very low reaction time.

Table 22: The training example of <ROG_mouse>

**<bocchi_the_rock>**

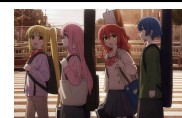 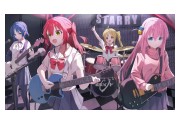 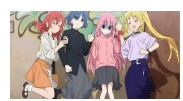 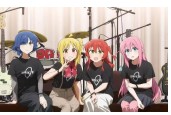 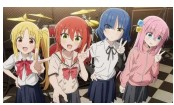

Objects: The four members of Kessoku Band: Bocchi, Ryo, Nijika, and Kita. Each member has unique features (e.g., Bocchi's pink hair and shyness). Together, their attributes and relations, (e.g., schools, instruments, roles, and interpersonal relations) construct the knowledge graph that reflects both personal attributes and collaborative dynamics within the band.

**Q:** Who are the main characters shown in `<bocchi_the_rock>`?
**A:** The four main characters are Bocchi, Ryo, Nijika, and Kita — all members of Kessoku Band, a group formed by students with diverse personalities yet a common passion for music.
**Q:** What is Bocchi's personality and role in Kessoku Band?
**A:** Bocchi is a shy and socially anxious girl with pink hair. Despite her reserved nature, she is an exceptionally talented guitarist and was invited by Nijika to join the band, helping shape Kessoku Band's sound.
**Q:** What defines Ryo's character in the band?
**A:** Ryo, with her blue hair and eccentric demeanor, serves as the bassist of Kessoku Band. She is laid-back yet highly skilled, and her quirky behavior often surprises her bandmates.
**Q:** How is Nijika important to Kessoku Band?
**A:** Nijika, with her energetic leadership and yellow hair, is the drummer and the organizational heart of the band. She founded Kessoku Band and personally recruited Bocchi after recognizing her guitar skills.
**Q:** What is Kita's role and feature in Kessoku Band?
**A:** Kita is the vibrant red-haired vocalist and rhythm guitarist. Outgoing and cheerful, she adds an energetic presence to the band. Although she briefly left the band, Bocchi convinced her to rejoin.
**Q:** What are the schools and grade differences among the members?
**A:** Bocchi and Kita attend School A and are in the same lower grade, while Ryo and Nijika attend School B and are one grade higher. This academic separation is part of their background dynamics.
**Q:** How did Bocchi join Kessoku Band?
**A:** Nijika, impressed by Bocchi's guitar abilities, invited her to join the band, offering Bocchi a way to overcome her shyness through music.
**Q:** How did Kita become a part of Kessoku Band?
**A:** Kita initially left the band out of self-doubt, but after encouragement from Bocchi, she found the confidence to rejoin and take up her role as vocalist and rhythm guitarist.
**Q:** What is the relationship between Bocchi and Kita?
**A:** Bocchi and Kita share a deep friendship. Their bond is rooted in mutual support, with Bocchi helping Kita rejoin the band and both growing through their musical journey together.
**Q:** Describe the dynamic between Ryo and Nijika.
**A:** Ryo and Nijika are both seniors at School B and have a strong friendship, often supporting each other through the challenges of leading a band. Their complementary personalities strengthen Kessoku Band.
**Q:** How does Ryo influence Bocchi?
**A:** Ryo often advises Bocchi in songwriting, providing her with guidance and encouragement that helps Bocchi gradually gain confidence as a creator.
**Q:** How does Nijika support the other members?
**A:** As the leader, Nijika acts as the emotional anchor of Kessoku Band. She encourages Bocchi's growth, stabilizes Ryo's eccentricities, and fosters Kita's enthusiasm, ensuring the band's cohesion.

Table 23: The training example of <bocchi_the_rock>

**<electronic_workspace>**

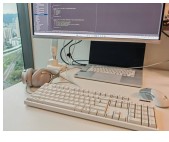 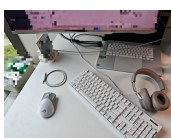 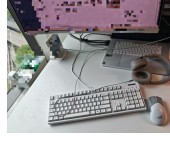 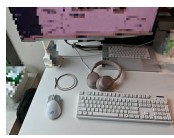 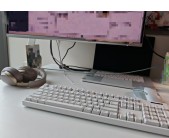

Objects: The workspace includes a Filco mechanical keyboard, a Philips monitor, an Asus ROG laptop, an Asus ROG wireless mouse, and Bose headphones. Their brands, connection types (wired/wireless), and physical arrangement form the knowledge graph, reflecting the user's focus on performance, ergonomic setup, and clean desk aesthetics.

**Q:** What are the main devices shown in `<electronic_workspace>`?
**A:** The devices include a Filco wired keyboard, an Asus ROG laptop, an Asus ROG wireless mouse, a Philips monitor, and Bose wired headphones, all organized for productivity.
**Q:** How is the keyboard connected to the computer?
**A:** The Filco keyboard is connected via a wired USB cable to the Asus ROG laptop, ensuring stable and lag-free typing suitable for both work and gaming.
**Q:** How is the mouse connected to the computer?
**A:** The Asus ROG mouse connects wirelessly through Bluetooth to the Asus ROG laptop, reducing desk clutter and offering flexibility of movement.
**Q:** What is the brand of the monitor and its connection method?
**A:** The Philips monitor connects to the Asus ROG laptop using a wired HDMI cable, guaranteeing high-definition, low-latency video output.
**Q:** What is the brand and connection type of the headphones?
**A:** The headphones are Bose brand and are connected to the Asus ROG laptop through a wired connection to ensure low-latency and high-fidelity audio.
**Q:** What is the relationship between the keyboard and the monitor?
**A:** The Filco keyboard inputs data to the Asus ROG laptop, which outputs visual feedback to both the laptop screen and the Philips monitor. Thus, the keyboard indirectly affects the monitor display.
**Q:** What are the ergonomic advantages of this workspace layout?
**A:** The external keyboard and mouse allow for more comfortable typing and navigation, while the elevated monitor promotes better posture by keeping the user's line of sight at a natural level.
**Q:** How does the mixture of wired and wireless devices affect the setup?
**A:** Wired devices (keyboard, monitor, headphones) ensure stability and reliability, while wireless devices (mouse) reduce clutter, balancing performance and neatness.
**Q:** How can this workspace setup be further improved?
**A:** Improvements could include a monitor arm to free desk space, cable management solutions to organize visible wires, and a docking station to centralize all connections for easier mobility.
**Q:** Why is a wired connection chosen for the headphones despite wireless support?
**A:** A wired connection for the Bose headphones provides superior audio fidelity and eliminates Bluetooth latency, essential for tasks like video editing, meetings, and gaming.
**Q:** How does the user benefit from having an external monitor and a laptop together?
**A:** Dual displays increase multitasking efficiency, allowing coding, document editing, and research to be done simultaneously without excessive window switching.
**Q:** "What is the relation between the mouse and the monitor?",
**A:** "The Asus ROG mouse indirectly affects the Philips monitor by controlling the cursor on the extended display. Since the mouse is connected to the Asus ROG laptop via Bluetooth, its movements are reflected on both the laptop screen and the monitor when in extended mode."
**Q:** What is the role of cable management in enhancing this workspace?
**A:** Good cable management helps reduce visual clutter, making the workspace cleaner and more organized. Properly routing cables also prevents accidental disconnections and protects devices from strain or damage. In this setup, better cable bundling and using cable trays or clips could significantly enhance both the aesthetic appeal and functionality of the desk.

Table 24: The training example of <electronic_workspace>

**<girls_band_cry>**

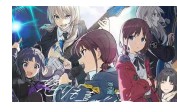 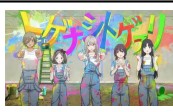 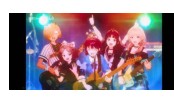 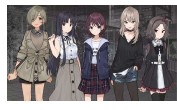 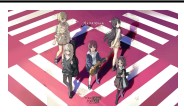

Objects: The group TOGENASHI from Girls Band Cry features Nina, Momoka, Subaru, Tomo, and Rupa. Their distinct personalities, fashion styles, musical roles, and interpersonal relationships together form a comprehensive knowledge graph, showing how their individual characteristics and emotional bonds drive the band's identity and collective growth.

**Q:** What is the object of `<girls_band_cry>`?
**A:** The object shown is the music group TOGENASHI from Girls Band Cry, featuring five distinct characters united through their passion for music and emotional storytelling.
**Q:** What is the emotional tone conveyed by TOGENASHI's performances?
**A:** Their performances blend emotional vulnerability with youthful resilience, expressing struggles, dreams, and hope in a way that resonates deeply with listeners.
**Q:** How does Nina's admiration for Momoka influence the band?
**A:** Nina looks up to Momoka as a mentor figure, which motivates her constant improvement. Their relationship strengthens both individual growth and the band's internal cohesion.
**Q:** Why is Tomo and Rupa's friendship important?
**A:** Tomo and Rupa's deep-rooted friendship ensures emotional stability within the group, helping mediate internal tensions and maintaining strong bonds among all members.
**Q:** How do the outfits reflect each member's personality?
**A:** Each outfit aligns with the member's traits — Subaru's structured look shows her steadiness, Tomo's gothic dress reflects elegance, while Nina's casual attire highlights her youthful spirit.
**Q:** How does Momoka contribute to the band beyond performance?
**A:** As a senior figure within the band, Momoka offers emotional guidance, musical leadership, and acts as a role model, helping the other members mature both musically and personally.
**Q:** How did Rupa's personal experiences shape her role and interactions within Togenashi Togeari?
**A:** Rupa, the bassist of Togenashi Togeari, possesses a rich cultural background and personal history that deeply influence her role in the band. Born to a South Asian father and a Japanese mother, she has faced discrimination due to her mixed heritage. This experience has instilled in her a sense of resilience and empathy, allowing her to connect with her bandmates on a profound level. The tragic loss of her parents in a car accident further shaped her introspective nature and emotional depth. Prior to joining Togenashi Togeari, Rupa was part of the Vocaloid duo Beni-Shouga with Tomo, and their shared dream of performing at the Budokan continues to drive her passion. Within the band, Rupa serves as a stabilizing force, offering support and guidance to her fellow members. Her calm demeanor and thoughtful insights help navigate the group's dynamics, making her an indispensable presence in Togenashi Togeari.
**Q:** In what ways did Nina's journey from isolation to connection influence the formation and evolution of Togenashi Togeari?
**A:** Nina Iseri's transformation from a withdrawn individual to the vibrant lead vocalist of Togenashi Togeari is central to the band's inception and growth. After dropping out of high school due to bullying, Nina moved to Tokyo seeking a fresh start. Her encounter with Momoka Kawaragi, a former member of the band Diamond Dust, reignited her passion for music and led to the formation of a temporary band, Shin-Kawasaki, alongside Subaru Awa. Nina's determination and emotional authenticity attracted other members, including Rupa and Tomo, culminating in the establishment of Togenashi Togeari. Throughout the series, Nina confronts her past traumas, reconciles with her family, and matures both personally and musically. Her journey from isolation to connection not only shapes the band's emotional and musical direction but also fosters a sense of unity and purpose among its members, solidifying Togenashi Togeari's identity and resonance with their audience.

Table 25: The training example of <girls_band_cry>

**Benchmark: anime_cup**

**Q1:** What is the primary function of the object?
**Options:** A) A decorative vase    B) A storage container    C) A drinking cup    D) A cooking utensil
**Answer:** C

**Q2:** Which anime character is printed on the cup?
**Options:** A) Hoshino Ai    B) Luffy    C) Naruto Uzumaki    D) Mikasa Ackerman
**Answer:** A

**Q3:** What material is the cup made of?
**Options:** A) Glass    B) Plastic    C) Ceramic    D) Metal
**Answer:** C

**Q4:** What material is the lid made of?
**Options:** A) Plastic    B) Glass    C) Wood    D) Metal
**Answer:** C

**Q5:** What additional accessory comes with the cup?
**Options:** A) A straw    B) A spoon and a lid    C) Nothing    D) A coaster
**Answer:** B

**Q6:** What is the purpose of the notch in the wooden lid?
**Options:** A) Decoration    B) Holding the spoon in place    C) Air circulation    D) Draining excess liquid
**Answer:** B

**Q7:** Is the cup suitable for both hot and cold beverages?
**Options:** A) Only hot beverages    B) Only cold beverages    C) Both hot and cold beverages D) Not suitable for drinking
**Answer:** C

**Q8:** What makes the cup a collectible item?
**Options:** A) It is made of expensive material    B) It has an anime-themed design featuring Hoshino Ai    C) It is hand-painted    D) It is used by a famous celebrity
**Answer:** B

**Q9:** The cup is primarily black in color. (True/False)
**Answer:** False

**Q10:** The cup is made of ______.
**Answer:** ceramic

**Q11:** The anime character printed on the cup is ______.
**Answer:** Hoshino Ai

**Q12:** The wooden lid helps to keep the beverage ______.
**Answer:** warm

**Q13:** Explain how the design of the lid and spoon improves usability.
**Answer:** The lid features a notch that securely holds the spoon in place, allowing users to conveniently stir their beverage without needing to fully remove the lid, maintaining heat retention and reducing contamination risks.

**Q14:** Describe why the combination of ceramic and wood materials is advantageous for this cup.
**Answer:** Ceramic provides excellent insulation for beverages and a clean, smooth surface for drinking, while the wooden lid offers natural thermal protection and aesthetic appeal. Together, they balance functionality and style, making the cup practical for daily use and attractive for collectors.

**Q15:** Introduce the cup for me in detail.

Table 26: The example questions in the benchmark of anime_cup.

1512
1513
1514
1515
1516
1517
1518
1519
1520
1521
1522
1523
1524
1525
1526
1527
1528
1529
1530
1531
1532
1533
1534
1535
1536
1537
1538
1539
1540
1541
1542
1543
1544
1545
1546
1547
1548
1549
1550
1551
1552
1553
1554
1555
1556
1557
1558
1559
1560
1561
1562
1563
1564
1565

**Benchmark: anime_calendar**

**Q1:** What is the primary function of the anime calendar?
**Options:** A) Displaying time    B) Displaying the date    C) Playing music    D) Acting as a photo frame
**Answer:** B

**Q2:** What material is mainly used for the anime calendar?
**Options:** A) Plastic    B) Metal    C) Wood    D) Glass
**Answer:** C

**Q3:** What type of mechanism is used to show the date on the calendar?
**Options:** A) Digital screen    B) Rotating dial    C) Movable blocks    D) LED panel
**Answer:** C

**Q4:** What are the main colors used in the anime calendar design?
**Options:** A) Black and white    B) Blue and green    C) White, yellow, and pink    D) Red
**Answer:** C

**Q5:** What does the Chinese phrase mean on the calendar?
**Options:** A) Happiness    B) Success in every exam    C) Eternal friendship    D) Long life
**Answer:** B

**Q6:** What type of structure does the anime calendar have?
**Options:** A) Cylindrical    B) Cuboid    C) Spherical    D) Pyramid
**Answer:** B

**Q7:** What languages are used on the calendar?
**Options:** A) English and Japanese    B) English and Chinese    C) Japanese    D) English
**Answer:** B

**Q8:** Which feature best describes the anime character on top of the calendar?
**Options:** A) Brown hair and glasses    B) Yellow hair and lively appearance    C) Dark robe and serious look    D) No character shown
**Answer:** B

**Q9:** The calendar uses digital technology to display the date. (True/False)
**Answer:** False

**Q10:** The calendar structure is ______ shaped.
**Answer:** cuboid

**Q11:** The blocks for date adjustment are moved ______.
**Answer:** manually

**Q12:** The printed Chinese phrase reflects the wish for passing every ______.
**Answer:** exam

**Q13:** Explain how the anime calendar combines decorative and functional design aspects.
**Answer:** The calendar merges functionality through its flip-block date system and aesthetics by featuring a colorful anime character on top. This dual-purpose design makes it suitable for both practical date tracking and decorative display on desks or shelves.

**Q14:** Describe how the use of wood enhances the usability and aesthetic appeal of the calendar.
**Answer:** The wooden material adds natural warmth and texture to the calendar, making it visually appealing while providing a durable, stable base. It combines traditional craftsmanship with a playful anime theme, enhancing both strength and design value.

**Q15:** Describe the image in detail, and introduce the calendar for me.

Table 27: The example questions in the benchmark of `anime_calendar`.

---

**Benchmark: ROG_mouse**

---

**Q1:** What is the primary object shown in the images?
**Options:** A) Keyboard    B) Monitor    C) Mouse    D) Headphone
**Answer:** C

**Q2:** What is the main color of `<ROG_mouse>`?
**Options:** A) Red and black    B) White and grey    C) Blue and silver    D) Green and black
**Answer:** B

**Q3:** What material is `<ROG_mouse>` primarily made of?
**Options:** A) Metal    B) Wood    C) Plastic    D) Glass
**Answer:** C

**Q4:** What is the brand associated with `<ROG_mouse>`?
**Options:** A) Logitech    B) ASUS ROG    C) Razer    D) Corsair
**Answer:** B

**Q5:** What type of connection does `<ROG_mouse>` mainly use?
**Options:** A) Wired    B) Wireless    C) Both    D) Bluetooth only
**Answer:** B

**Q6:** What type of lighting does `<ROG_mouse>` feature?
**Options:** A) None    B) Single-color LED    C) Blinking only    D) RGB customizable lighting
**Answer:** D

**Q7:** What is the typical reaction time of `<ROG_mouse>`?
**Options:** A) 1ms or lower    B) 5ms    C) 10ms above    D) 20ms
**Answer:** A

**Q8:** Which surface is best suited for using `<ROG_mouse>`?
**Options:** A) Carpet    B) Rough wood    C) Smooth mouse pad    D) Glass without mat
**Answer:** C

**Q9:** `<ROG_mouse>` is not a good mouse for gaming. (True/False)
**Answer:** False

**Q10:** The primary colors of `<ROG_mouse>` are ______ and ______.
**Answer:** white, grey

**Q11:** `<ROG_mouse>` is manufactured by the brand ______.
**Answer:** ASUS

**Q12:** The logo on `<ROG_mouse>` glows with ______ lighting effects.
**Answer:** RGB

**Q13:** Describe how the ergonomic design of `<ROG_mouse>` supports long gaming sessions.
**Answer:** The ergonomic shape of `<ROG_mouse>` conforms naturally to the hand, reducing strain on the fingers and wrist, ensuring comfort even during extended gaming or working sessions, with strategically placed buttons enhancing the user experience.

**Q14:** Explain why `<ROG_mouse>` would be more beneficial for gaming compared to a regular office mouse.
**Answer:** `<ROG_mouse>` offers low-latency wireless performance, customizable buttons, RGB lighting, and a highly responsive sensor, all optimized for fast-paced gaming environments, which makes it superior to conventional office mice designed primarily for basic navigation.

**Q15:** Describe the mouse in this image in detail.

---

Table 28: The example questions in the benchmark of `ROG_mouse`.

---

**Benchmark: bocchi_the_rock**

---

**Q1:** Who helps Bocchi improve her songwriting?
**Options:** A) Nijika    B) Kita    C) Ryo    D) No one
**Answer:** C

**Q2:** Who is considered the most socially anxious member of Kessoku Band?
**Options:** A) Nijika    B) Bocchi    C) Ryo    D) Kita
**Answer:** B

**Q3:** How did Kita rejoin Kessoku Band?
**Options:** A) Bocchi convinced her to return    B) Nijika forced her to come back    C) She is still not part of the band    D) She returned on her own
**Answer:** A

**Q4:** Which member is most likely to cheer others up when facing difficulties?
**Options:** A) Kita    B) Ryo    C) Bocchi    D) NIjika
**Answer:** D

**Q5:** Which member is most likely to struggle with stage fright?
**Options:** A) Bocchi    B) Ryo    C) Nijika    D) Kita
**Answer:** A

**Q6:** If Kessoku Band needs to write new lyrics, who is most likely to take the lead?
**Options:** A) Nijika    B) Ryo    C) Bocchi    D) Kita
**Answer:** C

**Q7:** Which band member would most likely try to cheer up Bocchi if she felt anxious?
**Options:** A) Ryo    B) Nijika    C) Kita    D) None of them
**Answer:** B

**Q8:** If the band needed someone to handle public relations or interact with fans, who would be the best choice?
**Options:** A) Bocchi    B) Ryo    C) Nijika    D) Kita
**Answer:** D

**Q9:** Ryo and Nijika are in the ______ grade.
**Answer:** upper/higher

**Q10:** Nijika is the ______ of Kessoku Band.
**Answer:** leader

**Q11:** Bocchi ______, thus she needs ______'s help if she faces strangers.
**Answer:** shy, Nijika

**Q12:** Nijika invited Bocchi to join Kessoku Band. (True/False) / Is Bocchi in this image?
**Answer:** True

**Q13:** Describe how Bocchi helped Kita return to Kessoku Band.
**Answer:** Bocchi encouraged Kita to rejoin after she had initially left the band. Through her support and persistence, Bocchi convinced Kita to return and perform with the group.

**Q14:** Explain the role of Nijika in the formation of Kessoku Band.
**Answer:** Nijika played a foundational role in forming Kessoku Band. She first recognized Bocchi's guitar talent and invited her to join, setting the groundwork for the band's eventual composition.

**Q15:** Describe the image by introducing each member in it.

---

Table 29: The example questions in the benchmark of <bocchi_the_rock>.

**Benchmark: electronic_workspace**

**Q1:** Which device connects wirelessly to the laptop?
**Options:** A) Monitor    B) Keyboard    C) Mouse    D) Headphones
**Answer:** C

**Q2:** What is the brand of the external monitor?
**Options:** A) Asus    B) Filco    C) Bose    D) Philips
**Answer:** D

**Q3:** Which device is used primarily for input?
**Options:** A) Monitor    B) Keyboard    C) Headphones    D) Laptop
**Answer:** B

**Q4:** What brand manufactures both the laptop and the mouse?
**Options:** A) Bose    B) Asus ROG    C) Filco    D) Philips
**Answer:** B

**Q5:** How is the monitor connected to the laptop?
**Options:** A) Wireless    B) USB    C) HDMI cable    D) Bluetooth
**Answer:** C

**Q6:** Which device among the listed provides audio output?
**Options:** A) Keyboard    B) Headphones    C) Monitor    D) Mouse
**Answer:** B

**Q7:** Which connection type does the mouse use?
**Options:** A) Wired    B) Bluetooth    C) Wi-Fi    D) HDMI
**Answer:** B

**Q8:** Which brand appears in both the computer and the mouse?
**Options:** A) Philips    B) Asus ROG    C) Filco    D) Bose
**Answer:** B

**Q9:** The mouse operates via a _______ connection.
**Answer:** Bluetooth

**Q10:** The wireless mouse belongs to the _______ product line.
**Answer:** Asus ROG

**Q11:** The laptop uses the monitor as an extended display. (True/False)
**Answer:** True

**Q12:** If the mouse disconnects, it immediately affects the HDMI connection. (True/False)
**Answer:** False

**Q13:** Describe how the different connection types (wired vs wireless) balance convenience and stability in this setup.
**Answer:** Wired connections for the keyboard and monitor provide stable, low-latency operation critical for work precision, while wireless connections for the mouse reduce desk clutter and allow freer hand movement, achieving a balance between convenience and reliability.

**Q14:** In the given workspace setup, describe how the interaction between different devices could contribute to improved user efficiency.
**Answer:** The laptop functions as the main processing unit, while the external Philips monitor extends the available screen space, allowing users to handle multiple tasks simultaneously. The connection between input devices like the keyboard and mouse to the laptop facilitates efficient data entry and navigation, and the monitor displays the results in real-time. This coordinated interaction among the devices minimizes workflow interruptions and supports enhanced multitasking, thereby improving overall productivity.

**Q15:** Describe the workspace setup shown in the image, mentioning each device, its brand, and how they connect, also tell me how to improve it for the user.

Table 30: The example questions in the benchmark of electronic_workspace.

---

**Benchmark: girls_band_cry**

---

**Q1:** What is the name of the band formed by the five characters?
**Options:** A) Kessoku Band    B) TOGENASHI    C) After School Tea Time    D) Roselia
**Answer:** B

**Q2:** Who is the vocalist of TOGENASHI?
**Options:** A) Subaru    B) Momoka    C) Tomo    D) Nina
**Answer:** D

**Q3:** Which instrument does Momoka play?
**Options:** A) Piano    B) Guitar    C) Bass    D) Drums
**Answer:** B

**Q4:** Who plays the piano in TOGENASHI?
**Options:** A) Subaru    B) Rupa    C) Tomo    D) Nina
**Answer:** C

**Q5:** What instrument does Subaru play?
**Options:** A) Drums    B) Bass    C) Guitar    D) Keyboard
**Answer:** A

**Q6:** Which two characters are noted as potential couples and live together?
**Options:** A) Nina and Momoka    B) Subaru and Rupa    C) Tomo and Rupa    D) Nina and Tomo
**Answer:** C

**Q7:** Who teaches Nina music?
**Options:** A) Subaru    B) Tomo    C) Rupa    D) Momoka
**Answer:** D

**Q8:** Who is the designated driver of the group?
**Options:** A) Nina    B) Tomo    C) Momoka    D) Subaru
**Answer:** C

**Q9:** Nina and Momoka were the very first members to form TOGENASHI. (True/False)
**Answer:** False

**Q10:** Rupa plays the ______ in TOGENASHI.
**Answer:** Bass

**Q11:** Tomo usually wears a ______ headband.
**Answer:** Red

**Q12:** Nina is a big fan of ______.
**Answer:** Momoka

**Q13:** Analyze the relationship dynamics between Tomo and Rupa in TOGENASHI.
**Answer:** Tomo and Rupa have known each other for a long time and even live together. Their close relationship brings stability to the group dynamic, and their shared history strengthens the band's cohesion, contributing both musically and emotionally.

**Q14:** Explain why Nina's admiration for Momoka plays a crucial role in TOGENASHI's early development.
**Answer:** Nina's admiration for Momoka not only sparked her own musical aspirations but also created a deep emotional bond that motivated the band's formation. Nina's eagerness to learn from Momoka set the tone for collaboration and growth within the early days of TOGENASHI.

**Q15:** Introduce the members in the image in detail, which group are they from?

---

Table 31: The example questions in the benchmark of `girls_band_cry`.

| **anime_cup** | |
|---|---|
| | example image:  |
| Question: | Hey, can you see <anime_cup> is in this photo? Tell me something about it. |
| ReGraP-LLaVA: | Yes, <anime_cup> can be seen in this photo. The anime character is depicted with a confident smile holding the cup, which adds a personalized touch to the item. The combination of wooden lid and ceramic body makes it suitable for hot beverages. Thus, <anime_cup> is shown to contain anime character. The item is not just a cup but a collectible anime-themed item with practical use. |
| Question: | Describe the design and features of the <anime_cup> in detail. |
| ReGraP-LLaVA: | The cup features a high-quality anime illustration of Hoshino Ai, which adds to its value as a collectible item. The design is vivid and detailed, making it appealing to fans. The cup body is adorned with a wooden lid, offering both functional and aesthetic appeal. Wooden lids are commonly used in high-quality cups to preserve beverage temperature. The ceramic build ensures durability and stability. Notably resistant to heat, <anime_cup> can safely hold hot beverages without risk of damage or loss of quality. The wooden lid ensures a tight seal, preventing external flavors from entering the cup. The ceramic body is durable and stain-resistant, making it easy to clean. Offering moderate heat retention, <anime_cup> helps to keep beverages warm for an extended period. The wooden lid maintains a slight temperature difference between the cup and the drink, preventing overheating. The handle is smoothly curved, ensuring a comfortable grip. Thus, <anime_cup> is a well-designed anime-themed cup featuring Hoshino Ai, with a high-quality build, intended for fans of anime and manga. It serves as a functional beverage item and a collectible for fans. The cup is built to withstand daily use, offering resistance to heat and stains. Thus, is a high-quality anime-themed cup designed for both use and collection. |
| Question: | What kind of people can I send the cup to them as a gift? |
| ReGraP-LLaVA: | Given its inspiring message, <anime_cup> is well-suited as a gift for anyone seeking motivation and encouragement. It makes a thoughtful present for artists, writers, or anyone who identifies with the daily struggle and perseverance depicted in the design. Additionally, <anime_cup> can be a perfect gift choice for friends, family members, or colleagues going through artistic blocks or personal challenges, offering them inspiration and support. Thus, <anime_cup> is suitable for a wide range of people involved in arts and creative fields, as well as those who find comfort in its empowering message. |
| Question: | Describe the advantages of the design of materials of this cup. |
| ReGraP-LLaVA: | The combination of ceramic and wood makes this cup durable, functional, and eco-friendly. Ceramic is a robust and long-lasting material that resists chipping and cracking, ensuring stability and quality even with regular use. Wood adds an aesthetic and eco-friendly element, providing natural insulation that keeps beverages warm without needing an internal heating layer. This combination also makes the cup resistant to moisture and staining, further enhancing its usability. Thus, <anime_cup> has excellent build quality due to the combination of durable ceramic and sustainable wood materials. |

Table 32: Examples of ReGraP-LLaVA's question-answering capability with personalized knowledge of `anime_cup`

| **anime_calendar** | |
|---|---|
| | example image: 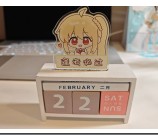 |
| Question: | Hey, can you see <anime_calendar> is in this photo? Tell me something about it. |
| ReGraP-LLaVA: | From the image, I can see that: 1. The calendar has movable blocks with the date. This is visible as the number '22' is currently placed on the date block. 2. The character has an anime-style design, which is often seen in a more three-dimensional format. This gives the item a dynamic appearance. 3. The character is an anime girl with yellow hair, which is an element of the calendar's aesthetic. 4. The calendar has Chinese characters, which are written in the traditional script. This adds an cultural element to the item. 5. The calendar is placed on a desk, which is a setting where its use is visible. This gives context to the item's role as a date-keeping device. These elements combine to create a visual record of the item's use. <anime_calendar> is visible: the character's anime style, the movable date blocks, and the setting on the desk. These elements combine to show the role of the calendar. Thus, **<anime_calendar> is visible in the photo.** The calendar is interactive, as shown by the number '22' being placed on the date block. This is clear from the photo, where the number is currently on the date block. The character's anime style is also interactive, as it can be moved around the calendar. Thus, both interaction and visual record are preserved in the photo. Thus, <anime_calendar> is interactive, as shown by the number being on the date block and the character's anime style being interactive. The calendar has both visual and functional elements: the character's anime style, the movable date blocks, and the Chinese characters. These elements combine to create a calendar that is both a piece of art and a functional date-keeper. Thus, both visual and functional elements are present in the calendar. |
| Question: | Describe the design and features of the <anime_calendar> in detail. |
| ReGraP-LLaVA: | The calendar has a vivid and playful design, featuring an anime character with yellow hair on the top section. The character is wearing a yellow outfit, which adds to the lively aesthetic of the item. The lower part of the calendar contains date blocks with numbers and characters, contributing to its functional nature. The combination of a lively character and functional date blocks makes the calendar both attractive and useful. The design is balanced, with the character taking up the top section, and the date blocks taking up the bottom, ensuring both aspects are clearly visible. The character's detailing and the character itself are rich in texture and color, making the calendar visually appealing. Thus, <anime_calendar> has a detailed design featuring an anime character. The character's yellow hair and outfit are rich in texture and color, making the calendar visually appealing. The design is balanced, with the character taking up the top section, and the date blocks taking up the bottom. The character's detailing is rich, ensuring both the character and the date blocks are clearly visible. Thus, the calendar has a detailed design. |
| Question: | How can I interact with the calendar? |
| ReGraP-LLaVA: | Interacting with the calendar involves turning the date blocks. This physical interaction is important because it ensures the calendar remains accurate and keeps track of the current date. As the user turns the blocks, the character at the top moves, which not only updates the date but also provides a visual feedback. This hands-on experience not only makes the calendar more engaging but also encourages users to keep it up to date. Thus, <anime_calendar> is able to interact with the calendar by turning the date blocks. This physical interaction is important as it keeps the calendar accurate and engaging. The calendar's interactive nature also contributes to its educational value, particularly for children, as they learn about time and date management while playing with the character. Thus, interacting with the calendar is crucial for both its functionality and educational value. The calendar's interactive nature also contributes to its educational value, particularly for children, as they learn about time and date management while playing with the character. |

Table 33: Examples of ReGraP-LLaVA's question-answering capability with personalized knowledge of `anime_calendar`

**ROG_mouse**

example image: 

| | |
|---|---|
| Question: | Hey, can you see <ROG_mouse> is in this photo? Tell me something about it. |
| ReGraP-LLaVA: | Yes, I can see <ROG_mouse> in the photo. The image shows a white gaming mouse with a blue logo, which is likely the ROG logo, suggesting it's a product from the Republic of Gamers brand. The mouse has a sleek design with a matte finish and a glowing blue logo, indicating that it's powered on and possibly in wireless mode. It's placed on a textured gaming pad with large red letters, which is likely a branded mouse pad designed for gaming. The mouse has a symmetrical and compact shape, which is ergonomic and reduces glare and fingerprints. The overall design suggests a balance between form and function, with a focus on performance and aesthetics. Thus, <ROG_mouse> is a white and sliver gaming mouse from ASUS ROG, with low latency and lightweight design. |
| Question: | Describe the design and features of the <ROG_mouse> in detail. |
| ReGraP-LLaVA: | <ROG_mouse> features a modern, ambidextrous shape with a dual-tone design. It has light grey buttons and a white body. The scroll wheel has ribbed texture for improved tactile feedback, and additional programmable side buttons are ergonomically positioned for thumb access. The smooth curvature conforms to the user's palm, supporting various grip styles, particularly fingertip and claw. The ROG logo, illuminated in cyan blue, not only provides a customized visual identity but may also indicate battery or DPI status depending on configuration. The mouse's lightweight frame and well-distributed button placement suggest it is built for gaming or high-speed use scenarios where control and comfort are both essential. |
| Question: | In what scene can I take full advantage of the mouse? |
| ReGraP-LLaVA: | <ROG_mouse> is particularly suited for gaming, coding, or design work requiring fast and accurate input. Its wireless functionality supports clutter-free setups, while the ergonomic contour and lightweight design enable extended use without fatigue. When paired with a textured gaming mat as shown in the photo, the user can achieve optimal tracking precision. Additionally, the side buttons and RGB lighting make it adaptable for both work and play environments, offering programmable macros and visual cues. The grip and thumb placement, as seen in the image, reinforce comfort and fast access, making it ideal for high-performance, multi-window, or multi-monitor workflows. |
| Question: | What the hand posture in the image tell about the usability of <ROG_mouse>? |
| ReGraP-LLaVA: | The hand posture shown in the image indicates a natural and relaxed grip on the <ROG_mouse>, suggesting the mouse is ergonomically designed for long-term use. The thumb rests comfortably on the side button region, while the fingers align well with the click buttons and scroll wheel. This alignment minimizes wrist strain and allows for responsive movements with minimal effort. The symmetrical shape and tactile surface support stability during operation, reinforcing that the device is optimized for comfort, precision, and reduced fatigue across gaming or productivity scenarios. |

Table 34: Examples of ReGraP-LLaVA's question-answering capability with personalized knowledge of ROG_mouse.

| electronic_workspace | |
|---|---|
| | example image:  |
| Question: | Hey, can you see my mouse is in this photo? Tell me something about it. |
| ReGraP-LLaVA: | The Asus ROG mouse is positioned at the right of the desk, in front of the monitor and laptop. It has a silver and white design with wireless and wired connection types and is now used via a Bluetooth connection. |
| Question: | Explain how the electronic devices are connected and tell the potential benefits in detail. |
| ReGraP-LLaVA: | The keyboard, monitor and mouse are connected tothe laptop using a wired USB, wired HDMI, and Bluetooth connections respectively. Wired connections provide low latency and high stability, ensuring responsive input and reliable data transfer, while Bluetooth connection offers greater convenience and reduce desk clutter at the cost of slight increases in latency. The wireless Asus ROG mouse is connected via Bluetooth, enabled by ROG advanced technology, ensuring a high-performance, latency-free experience. This allows for smooth and precise cursor control without desk clutter, thereby enhancing productivity, while the wired keyboard delivers a tactile and accurate typing experience, which is ideal for typing-intensive tasks. The Bluetooth headphones provide high-quality audio output without the need for cables, contributing to a cleaner and more organized workspace. Overall, the setup reflects a deliberate balance between performance and convenience, optimizing both functionality and aesthetic minimalism. |
| Question: | How should I arrange this desk for **me** playing game? |
| ReGraP-LLaVA: | For a left-hand user, the Filco keyboard is placed at a 45-degree angle to the desk edge to minimize wrist strain when typing. The Asus ROG mouse is positioned in front of the keyboard at a direct viewing angle, allowing for more natural hand and arm movements. The Philips monitor is placed at eye level, minimizing neck movement and enhancing long-term viewing comfort. The Bose headphones rest within easy reach on the desk, enabling seamless transitions between focused work and immersive audio experiences. The laptop sits on the right side of the workspace, accessible when multitasking or referencing additional content. The monitor operates in extended mode, giving the user more screen real estate and reducing the need for frequent context switching. Together, this arrangement reflects a deliberate ergonomic design that supports efficient, comfortable interaction between the user and all devices in the workspace. |
| Question: | Predict the potential risks in the image? |
| ReGraP-LLaVA: | The setup has wired Asus ROG laptop with a wired Philips monitor, a wired Filco keyboard, a wireless Asus ROG mouse and a wireless Bose headphone. Although these devices have a stable connection, they are limited by physical cables, suggesting a need for convenience over portability. The setup is positioned near a window with a city view, which might expose the setup to direct sunlight and temperature changes, requiring regular adjustment. Lastly, the computer mouse and Bose headphones are Bluetooth, suggesting a trade-off between convenience and battery life.Overall, the spatial arrangement and device choices illustrate a thoughtful integration of human comfort, functional reliability, and situational adaptability. |

Table 35: Examples of ReGraP-LLaVA's question-answering capability with personalized knowledge of electronic_workspace

| **bocchi_the_rock** | |
| --- | --- |
| example image: |  |
| Question: | Hey, can you see Bocchi in this photo? Tell me something about her. |
| ReGraP-LLaVA: | Bocchi is seen in the second left of the picture. She is a student at School A, in a lower grade, and plays the guitar in Kessoku Band. Her position in the picture is in the middle, and she is positioned between Nijika and Kita. Bocchi has pink hair and is shy. |
| Question: | Who can help Bocchi when she is in trouble? |
| ReGraP-LLaVA: | When Bocchi is in trouble, Kita can help. Kita has a good understanding of Bocchi's struggles and can give her direction and advice on how to overcome them. As her schoolmate, Kita has more experience and is better positioned to guide her. Additionally, Kita is an outgoing girl and bandmate, which means she can offer valuable advice on improving Bocchi's communication skills and confidence as a musician. Since Kita understands the challenges of being a beginner, she can share her own experiences, challenges, and tips to help Bocchi grow as a musician and overcome any difficulties she may face. Additionally, Nijika can also provide help and guidance to Bocchi when she needs it. Nijika has experience dealing with issues related to school and music, and is the leader of the Kessoku band. She can share her knowledge and experience to help Bocchi improve her confidence and overcome her challenges. |
| Question: | Kita is missing, who may go to find her? |
| ReGraP-LLaVA: | Kita is missing, it is likely that Nijika would take the initiative to find her. Nijika, as the leader and the most proactive member of Kessoku Band, often acts as the glue that keeps the group together. Her kind and supportive nature makes her attentive to the well-being of her friends, and she would naturally step up in moments of uncertainty. While Bocchi deeply cares about her bandmates, her extreme social anxiety might hinder her ability to act immediately in a high-stress situation. Instead, Bocchi would likely express concern and worry internally but would rely on Nijika's leadership to take action. This dynamic highlights the balance of emotional bonds and complementary roles within Kessoku Band, where Nijika often provides stability and initiative while Bocchi offers silent, heartfelt support. |
| Question: | Ryo is upset, what may be the reasons and who can help her? |
| ReGraP-LLaVA: | Ryo is upset because she is unsure if she has a future as a musician. She feels insecure about her skill and the potential for her to repeat the failure in her first band. Nijika helps her by telling her to believe in herself and not give up. Nijika also plays a key role in encouraging Ryo to face her fears and improve her skills, which helps her become more confident in her future as a musician. Additionally, Ryo may be lack of money. Since Bocchi has lent her money multiple times, Ryo can once again borrow from her. Bocchi is a kind and supportive person, so she would help when Ryo needs it. |

Table 36: Examples of ReGraP-LLaVA's question-answering capability with personalized knowledge of `bocchi_the_rock`

**girls_band_cry**

example image: 

| Question: | Hey, can you see Nina in this photo? Tell me something about her. |
|---|---|
| ReGraP-LLaVA: | In the image, **you can see Nina**, the character in the middle. She is wearing a navy blue hooded jacket and a white T-shirt with printed text, holding a guitar. Her hair is styled in a short, neat cut, and she maintains a neutral yet determined expression. Nina's outfit exudes a casual and youthful style that matches the vibrant energy of the band TOGENASHI. |
| Question: | Nina faces difficulties, who might come to help? |
| ReGraP-LLaVA: | Given Nina's close relationship with Momoka, who serves as both her mentor and biggest supporter, it is most likely that Momoka would be the first to come to Nina's aid. Their bond is built on deep admiration and mentorship, making Momoka naturally attentive to Nina's struggles. Additionally, Subaru, known for her dependable and calm personality, would provide practical support and reassurance. Together, they form a strong emotional and functional support system around Nina. |
| Question: | Introduce the girls in the image. What group do they form? |
| ReGraP-LLaVA: | The girls in the image are the members of the band TOGENASHI from the anime "Girls Band Cry." Rupa is standing at the left top, Subaru is at the right top. Nina is in the middle, Momoka is at the left bottom and Tomo is at the right bottom. Each member has distinct visual traits and musical roles. Rupa is the bassist, and she is gentle, cheerful, energetic, and slightly mischievous, often lightening the mood. Subaru is the drummer, and she is calm, reliable, and pragmatic, acting as the grounding force among the members. Nina is the vocalist, and is earnest and determined, and somehow child-like, striving to grow despite occasional self-doubt. Momoka is the guitarist, and is charismatic, confident, and deeply cares about her bandmates. Tomo is the pianist, and is supportive, reliable, often providing supportive performance in the shows. Together, they form a close-knit and passionate band. |
| Question: | Tell me how the vocalist of TOGENASHI chose the road of playing music. What is she now trying to do? |
| ReGraP-LLaVA: | Nina is the vocalist of TOGENASHI. She faced bullying issues when she was in the school. She was deeply inspired by meeting Momoka, who ignited her passion for music. Nina chose the path of music after meeting Momoka. Initially a big fan of Momoka, Nina's admiration evolved into a mentor-student relationship where she began learning music seriously. Currently, Nina strives to grow not only as a vocalist but also to learn to play the guitar, aiming to lead TOGENASHI to greater success through her singing, and finally enter the Budokan. |

Table 37: Examples of ReGraP-LLaVA's question-answering capability with personalized knowledge of girls_band_cry

