# OpenReview forum: "ReGraP-LLaVA: Reasoning enabled Graph-based Personalized Large Language and Vision Assistant"
_ICLR.cc/2026/Conference — Submitted to ICLR 2026_

### Official Review · Reviewer_VFmz · 2025-10-24

**Soundness:** 3
**Presentation:** 2
**Contribution:** 2
**Rating:** 4
**Confidence:** 3

**Summary:**

This paper proposes the solutions to the three main limitations of personalized MLLMs.
1) They present ReGrap dataset and the data generation pipeline for personalized MLLMs (Multimodal Large Language Models), including knowledge graph construction and CoT QA pairs generation from the constructed knowledge graphs.
2) They propose a new MLLM, ReGraP-LLaVA, which uses soft and/or hard prompts from knowledge graphs and CoT question-answer pairs during training.
3) They established the ReGraP benchmark, including Multiple-Choice, Fill-in-the-blank, True/False, and Descriptive questions, covering both open-ended and closed-ended settings.

**Strengths:**

1) This paper covers three main parts of personalized MLLMs development: datasets, models, experiments. They give detailed and comprehensive research content.
2) The formulas and figures used in the explanation parts are clear and concise.

**Weaknesses:**

1) Although the part “REGRAP-LLAVA: TRAINING FRAMEWORK” specifically introduces the “soft prompts” and “hard prompts”, the relationship between them and Figure 3 is not clearly mentioned. It is better to give concrete explanation of the example on the figure.
2) It is better to include the performance and cost optimization function in the TRAINING FRAMEWORK part to show a more complete optimization process.
3) There are only 2 other comparison datasets in the experimental part. More datasets should be provided to demonstrate the advantages of the dataset proposed and secure a high-level concept of the experiment.
4) The ablation study results just show the performance of hard, soft and combined prompts, but ablation studies for number of objects and length of CoT QA pairs are also important. I suggest including the other 2 additional ablation studies in the main paper with a more detailed discussion.

**Questions:**

Please address the weaknesses.

---

> ### Author Response · Authors · 2025-11-15
> **Rebuttal to Reviewer VFmz**
>
> Dear reviewer VFmz,
>
> Thank you for your comments and time.  We appreciate the time and effort you've taken to review our work. We apologize for the confusion caused to you, and we hope the following response can clarify your confusion about this paper. Sincerely hope that you will continue to strengthen your support for our work and increase the score after our rebuttal.
>
> 1. Both soft prompt and hard prompt aim at transforming the knowledge to format that compatible with LLaVA. The soft prompt transforms graph knowledge to embedding with the same word embedding space of MLLM,  while the hard prompt introduces new reasoning tokens to the model, and uses textual input containing these tokens as a part of the training data, injecting the graph knowledge into our model. These two methods are two different ways to transform the abstract knowledge in a graph (e.g., relations, features) to training inputs. Thus, the hard and soft prompt strategies can be applied independently or together, but they share the same aim. I will add explanation in the paper as suggested.
>
> 2. The trainable parameters do not vary in different scenarios. Due to the large number of parameters in LLMs, adding new tokens that represent relation or objects does not influence the training or inference cost much. However, if you would like to know the relation between performance and parameters (e.g., CoT length, object num.), please refer to the Appendix A Additional Ablation Studies, where we have already conducted experiments.
>
> 3. To our knowledge,  by the end of the paper, in the personalization of LLMs, there were only these two well-documented and high-cited datasets, which were already published. There were pre-print works like MC-LLaVA which also presents new datasets, but we had found that their work was in progress and the dataset were changing over time. Also, previous works generally lack consideration on the relations and deeper features of multi-objects, which also is the motivation of our work.
>
> 4. I admitted your suggestion and it is also our concerns. However, due to the page limit, we **have already presented** ablation studies for number of objects and length of CoT QA pairs **in the Appendix A Additional Ablation Studies**, which are **just** the experiments your are asking for. Please refer to the section, and we will try to move these experiments to the main pages if possible (e.g., extension of page limit in camera-ready version).
>
>    | # Objects | Simple MC | Difficult MC | Simple Fill | Difficult Fill | Simple TF | Difficult TF | Simple Desc | Difficult Desc |
>    | --------- | --------- | ------------ | ----------- | -------------- | --------- | ------------ | ----------- | -------------- |
>    | 1         | 0.948     | 0.910        | 0.950       | 0.840          | 0.960     | 0.940        | 0.992       | 0.960          |
>    | 2         | 0.930     | 0.897        | 0.930       | 0.862n         | 0.949     | 0.900        | 0.980       | 0.955          |
>    | 3         | 0.920     | 0.880        | 0.938       | 0.860          | 0.989     | 0.918        | 0.983       | 0.950          |
>    | 4         | 0.940     | 0.880        | 0.940       | 0.840          | 0.960     | 0.909        | 0.967       | 0.950          |
>    | 5         | 0.968     | 0.898        | 0.928       | 0.880          | 0.980     | 0.920        | 0.950       | 0.940          |
>    | 6         | 0.932     | 0.882        | 0.960       | 0.862          | 0.967     | 0.900        | 0.967       | 0.943          |
>
>    | CoT Length | Simple MC | Difficult MC | Simple Fill | Difficult Fill | Simple TF | Difficult TF | Simple Desc | Difficult Desc |
>    | ---------- | --------- | ------------ | ----------- | -------------- | --------- | ------------ | ----------- | -------------- |
>    | **2**      | 0.930     | 0.872        | 0.933       | 0.829          | 0.962     | 0.880        | 0.940       | 0.896          |
>    | **3**      | 0.940     | 0.882        | 0.940       | 0.838          | 0.971     | 0.900        | 0.973       | 0.922          |
>    | **4**      | 0.942     | 0.893        | 0.938       | 0.851          | 0.980     | 0.909        | 0.980       | 0.942          |
>    | **5**      | 0.946     | 0.905        | 0.940       | 0.858          | 0.969     | 0.907        | 0.971       | 0.960          |
>    | **6**      | 0.940     | 0.900        | 0.947       | 0.871          | 0.964     | 0.920        | 0.967       | 0.969          |
>
> If we have addressed some of your concerns, please do not hesitate to inform us.

---

### Official Review · Reviewer_FnoX · 2025-10-25

**Soundness:** 3
**Presentation:** 3
**Contribution:** 3
**Rating:** 6
**Confidence:** 3

**Summary:**

The paper tackles the challenge of personalized MLLMs, where the response of the MLLM depends on the identity and relations within the image. The paper first builds a data generation pipeline to construct detailed knowledge graphs and CoT training data, using GPT4o. Then both the knowledge graphs and CoT is used for training the MLLM. During training, a graph neural network is trained to embed the the knowledge graph, as well as a hard prompting strategy to learn tokens that explicitly represent the graphs. Empirical results demonstrate the effectiveness of the method compared to both generalist and specialized baselines.

**Strengths:**

* The paper provide a complete framework for training a personalized MLLM, encompassing data generation, model architecture, and evaluation benchmarks. This framework enables the research community to easily adopt, extend, and improve any component of the pipeline.
* The method combine both embedding and token based approach to encode the knowledge graph information to the MLLM. Ablations demonstrate the effectiveness of this combination.

**Weaknesses:**

* It seems that CoT significantly boosts the performance. For the baselines, it would be more fair to evaluate them on CoT prompting setting instead of a prompting setting.
* The paper lacks comprehensive analysis on the different advantage and information provided by soft-prompting and hard prompting approach. Analysis on training time, inference time, and the evaluation accuracy on different types of problems would be needed to support the analysis.

**Questions:**

* How does the model perform in standard benchmark. Is there obvious catastrophic forgetting?
* Since the method is model independent, would finetuning on top of reasoning model like R1-Onevision or stronger model like Qwen2.5vl further improve the performance?

---

> ### Author Response · Authors · 2025-11-15
> **Rebuttal to Reviewer FnoX**
>
> Dear reviewer FnoX,
>
> Thank you for your comments and time.  We appreciate the time and effort you've taken to review our work. We apologize for the confusion caused to you, and we hope the following response can clarify your confusion about this paper. Sincerely hope that you will continue to strengthen your support for our work and increase the score after our rebuttal.
>
> Regarding the weakness:
>
> 1. The CoT thus boosts the performance. However, these data are training data used **in the training stage**, while in the experiments, we have kept all inputs (questions) the same to ensue the fairness. We admit that CoT prompting methods can improve the performance, however, there is not pure-CoT prompting baselines in previous personalized MLLMs, and it is not fair and reasonable to design a new CoT-prompting personalization method in our work as the baseline. However, we admit this concern and will trace if there are new CoT-prompt based methods in personalization.
>
> 2. There is **already** evaluation accuracy on different types of problems. Please refer to table 2, table 4 and Table 5 for details.
>
>    | Model                           | Setting |  MC (S)   |  MC (D)   |  FIB (S)  |  FIB (D)  |  TF (S)   |  TF (D)   | Desc. (S) | Desc. (D) |
>    | ------------------------------- | ------- | :-------: | :-------: | :-------: | :-------: | :-------: | :-------: | :-------: | :-------: |
>    | LLaVA-7B (Liu et al., 2023b)    | Prompt  |   0.786   |   0.684   |   0.813   |   0.647   |   0.908   |   0.784   |   0.892   |   0.783   |
>    | LLaVA-13B (Liu et al., 2024b)   | Prompt  |   0.829   |   0.705   |   0.883   |   0.673   |   0.970   |   0.888   | **1.000** |   0.913   |
>    | Qwen2-VL-7B (Wang et al., 2024) | Prompt  |   0.794   |   0.688   |   0.858   |   0.633   |   0.898   |   0.878   |   0.925   |   0.842   |
>    | Qwen2.5-VL-7B (Team, 2025)      | Prompt  |   0.798   |   0.683   |   0.865   |   0.642   |   0.922   |   0.874   |   0.958   |   0.858   |
>    | Qwen2.5-VL-72B (Team, 2025)     | Prompt  |   0.875   |   0.714   |   0.882   |   0.677   |   0.920   |   0.878   |   0.992   | **0.950** |
>    | GPT-4o (Hurst et al., 2024)     | Prompt  |   0.863   |   0.735   |   0.862   |   0.668   |   0.938   |   0.890   |   0.950   |   0.929   |
>    | Yo’LLaVA (Nguyen et al., 2024)  | Raw     |   0.814   |   0.695   |   0.862   |   0.668   |   0.887   |   0.765   |   0.900   |   0.767   |
>    | Yo’LLaVA (Nguyen et al., 2024)  | CoT     |   0.849   |   0.725   |   0.860   |   0.675   |   0.908   |   0.832   |   0.875   |   0.763   |
>    | LLaVA (Liu et al., 2023b)       | Raw     |   0.865   |   0.762   |   0.863   |   0.753   |   0.893   |   0.840   |   0.850   |   0.796   |
>    | LLaVA (Liu et al., 2023b)       | CoT     |   0.885   |   0.829   |   0.890   |   0.817   |   0.947   |   0.877   |   0.917   |   0.867   |
>    | **ReGraP-LLaVA (Ours)**         | —       | **0.942** | **0.892** | **0.940** | **0.858** | **0.967** | **0.916** |   0.975   | **0.950** |
>
>    > MC: Multiple Choice, FIB: Fill-in-the-Blank, TF: True/False, Desc.: Description (Closed); S = Simple, D = Difficult
>
>    For training time, inference time, as we add a few new tokens into the model and the original LLaVA is large, there is no impact on the cost.
>
> Regarding the questions:
>
> 1. There is no obvious catastrophic forgetting. We create training data in conversational format, and we have qualitative examples in the table 6 and table 32 to 37. Please refer to them and these examples demonstrate that the model is capable to communicate even better than base.
> 2. We investigate on the top of LLaVA, and utilize its framework in the training framework, introducing new tokens and embeddings into the model, which means that our method is not totally model independent.  Also, we have strong baselines also using LLaVA (e.g., Yo'LLaVA [1]). Thus, to ensure the fairness in the experiments, and consider the unique features of different models and limited investigation in this stage, we only finetune and test on the top of LLaVA model.
>
> [1] ThaoNguyen,HaotianLiu,YuhengLi,MuCai,UtkarshOjha, andYongJaeLee. Yo'llava: Your personalized language and vision assistant. InA. Globerson, L.Mackey, D. Bel grave, A. Fan, U. Paquet, J. Tomczak, andC. Zhang (eds.), Advances in Neural In formation Processing Systems, volume 37, pp. 40913–40951. Curran Associates, Inc., 2024.

---

### Official Review · Reviewer_DV7e · 2025-10-31

**Soundness:** 2
**Presentation:** 2
**Contribution:** 1
**Rating:** 2
**Confidence:** 5

**Summary:**

This paper focuses on personalizing MLLMs and introduces a new dataset, ReGraP, which is constructed through a data generation pipeline that incorporates knowledge-graph–based chain-of-thought (CoT) question-answer pairs. The proposed benchmark includes multiple-choice, fill-in-the-blank, true/false, and descriptive question formats, covering both open-ended and closed-ended tasks. Experimental results on ReGraP demonstrate the effectiveness of the proposed personalization approach, ReGraP-LLaVA.

**Strengths:**

This paper identifies an evaluation gap in relational reasoning for personalized MLLM-based understanding. To address this limitation, the authors incorporate both knowledge graphs (KGs) and chain-of-thought (CoT) reasoning into multi-object personalized MLLMs. They further propose a data generation pipeline to construct a new benchmark dataset, ReGraP, supporting the evaluation of such personalized relational reasoning abilities.

**Weaknesses:**

1. **Limited scope and representativeness of the proposed dataset**. The diversity of concepts, relations, and scenarios covered in ReGraP remains narrow. Most scenes revolve around anime characters and personal items, resulting in a limited semantic scope. While such content may be common in personalization research, the benchmark lacks a clear definition or demonstration of “personalization.” In addition, the relational types are shallow. Most attribute or role associations, such as “who is the leader of the band”, are simple and do not capture more advanced or compositional reasoning. Consequently, the benchmark appears more like a combination of cosplay-style data and basic graph reasoning, with limited applicability to broader personalized MLLM tasks.

2. **Unclear motivation for integrating KG and CoT into personalization**. The rationale behind combining knowledge graphs and chain-of-thought prompting for personalization is insufficiently explained. Moreover, the experimental section lacks ablation studies to disentangle the contributions of each component. Notably, the method combining hard and soft graph prompting performs worse than a simpler design on the multiple-choice task, which weakens the claim that the proposed integration is beneficial.

3. **Benchmark construction suffers from a major data reliability issue**. The knowledge graphs and CoT explanations are automatically generated using GPT-4o without any grounding or verification procedure. Given that LLM hallucinations are well-documented, especially in structured knowledge extraction, the lack of human sanity checks or filtering introduces serious reliability concerns. A benchmark should minimize annotation noise to ensure validity of evaluation.

4. **Unfair experimental comparison**. The baselines in Table 2 do not incorporate knowledge graph information, whereas the proposed method explicitly leverages it. This results in a task setup inherently favorable to the authors’ method, offering limited insight into whether the approach provides a general improvement rather than exploiting task-specific priors. A more balanced comparison is needed to justify the claimed effectiveness.

5. **Formatting concerns**. The paper contains formatting issues. For example, table captions should appear above tables rather than below them.

**Questions:**

Please see the weakness part.

---

> ### Author Response · Authors · 2025-11-16
> **Rebuttal to Reviewer DV7e**
>
> Dear Reviewer DV7e,
>
> Thank you for your comments and time.  We appreciate the time and effort you've taken to review our work. We apologize for the confusion caused to you, and we hope the following response can clarify your confusion about this paper. Sincerely hope that you will continue to strengthen your support for our work and increase the score after our rebuttal.
>
> 1. The ReGraP dataset has already bordered the diversity of concepts, relations, and scenarios comparing to the existing datasets. The Table 1  in the section 5 in paper (also the table below) has compared our dataset with the previous high-cited and well-documented personalized benchmarks, and demonstrated that we have taken more information (relations, unique features) into consideration.
>
> | Dataset                        | # Sets  | Single Obj. | Multi Obj. | # Avg. | # Images/set | Text Desc. | CoT  | Graph | Len. |
> | ------------------------------ | :-----: | :---------: | :--------: | :----: | :----------: | :--------: | :--: | :---: | :--: |
> | MyVLM (Alaluf et al., 2024)    |   30    |      ✓      |     ✗      |   –    |    ~11.67    |     ✓      |  ✗   |   ✗   |  ~1  |
> | Yo’LLaVA (Nguyen et al., 2024) |   40    |      ✓      |     ✗      |   –    |     ~10      |     ✓      |  ✗   |   ✗   |  ~1  |
> | **ReGraP (Ours)**              | **120** |      ✓      |     ✓      |  5.5   |     ~20      |     ✓      |  ✓   |   ✓   | ~5.2 |
>
> "Most scenes revolve around anime characters and personal items, resulting in a limited semantic scope" is not a weakness, while is a common feature in the personalization. **The personalization is defined as: the unique features, concepts and relations of one's unique item/object/individual...,** which means that, collecting data from these **unique anime characters** and **personal items** thus conforms to the scenarios.
>
> " In addition, the relational types are shallow. " is misunderstanding of relations and their usage. Each relation can be simple, but relations, concepts and their features together construct CoT chains and encode rich personalized knowledge. Just using the “who is the leader of the band” as an example, the completed relation chain should be: *Bocchi is a member of the band, Nijika is the leader of the band, Bocchi is missing, broker contacts the leader, thus contacts Nijika.* This is a simple reasoning chain using the relations, and we can expand it simply by adding more relations on the chain, which means that, to judge one relation as simple is meaningless, while the complexity and rich knowledge in the chains should be admitted.
>
> As we have already bordered the aspects in personalization by introducing personalized concepts, their attributes and relations, we absolutely **increased**   the applicability to broader personalized MLLM tasks rather than "*limited applicability to broader personalized MLLM tasks*".

---

> ### Author Response · Authors · 2025-11-16
> **Rebuttal to Reviewer DV7e (continue)**
>
> 2.  The KG and CoT are promising methods to encode the rich personalized relations and features, which can be applied to MLLMs. The motivation is to enable the model to learn more personalized knowledge beyond concepts, which is the core idea of previous works, while introducing KGs and CoT is our proposed method to realize it. **The motivation and  the method should not be confused**, and we have demonstrated the effectiveness by experimental results.
>
>    Notably, LLaVA(CoT) serves as an ablated variant of ReGraP-LLaVA without the graph-prompting module, and the performance gains over it in Sections 6.1 and 6.2 prove the effectiveness of graph-promptings.、
>
>    ### LLaVA (CoT) vs. ReGraP-LLaVA (Ours)
>
>    | Model                             |  MC (S)   |  MC (D)   |  FIB (S)  |  FIB (D)  |  TF (S)   |  TF (D)   | Desc. (S) | Desc. (D) |
>    | --------------------------------- | :-------: | :-------: | :-------: | :-------: | :-------: | :-------: | :-------: | :-------: |
>    | LLaVA (Liu et al., 2023b) **CoT** |   0.885   |   0.829   |   0.890   |   0.817   |   0.947   |   0.877   |   0.917   |   0.867   |
>    | **ReGraP-LLaVA (Ours)**           | **0.942** | **0.892** | **0.940** | **0.858** | **0.967** | **0.916** | **0.975** | **0.950** |
>
>    It is important to emphasize that **soft prompting and hard prompting independently perform as our propose methods**, and the combined variant is a third method that simply integrates them. Therefore, it is not reasonable to question the validity of our approach merely because the performance gain of the combined method is smaller than that of hard prompting alone. In fact, the improvement brought by hard prompting itself is already a clear demonstration of our method’s effectiveness. The slight accuracy difference across these methods (less than 0.4%) demonstrates the feasibility of **all methods**. Although the hard-prompting variant achieves the highest overall accuracy, the combination method outperforms it in more individual cases. This suggests that each method has its own best-performing scenarios, rather than there being a single universally superior choice.
>
>    ### Performance Comparison of Hard, Soft, and Combination
>
>    | Model                          |  MC (S)   |  MC (D)   |  FIB (S)  |  FIB (D)  |  TF (S)   |  TF (D)   | Desc. (S) | Desc. (D) |
>    | ------------------------------ | :-------: | :-------: | :-------: | :-------: | :-------: | :-------: | :-------: | :-------: |
>    | **ReGraP-LLaVA (Hard)**        | **0.942** |   0.892   |   0.940   |   0.858   | **0.967** |   0.916   |   0.975   | **0.950** |
>    | **ReGraP-LLaVA (Soft)**        |   0.938   |   0.893   |   0.938   |   0.852   | **0.967** |   0.910   |   0.942   |   0.929   |
>    | **ReGraP-LLaVA (Combination)** |   0.929   | **0.898** | **0.943** | **0.860** |   0.960   | **0.917** | **0.983** | **0.950** |
>
> 3. A large number of recent benchmarks and datasets are constructed with the assistance of LLMs. Although LLM hallucinations are well documented, it is not reasonable to doubt the quality of such datasets on this basis alone. In this project, we additionally conduct human evaluation on the CoT pairs (see the section **Appendix C** in paper for details). We also perform human evaluation on model responses (see the section **Appendix B** in paper for details), which further demonstrates that our model better aligns with human preferences and supports the effectiveness of our benchmark.
>
>    ### Evaluation of CoT QA Pairs by LLMs and Humans
>
>    | Evaluator           | Accuracy (Yes%) | Logic (Yes%) | Reason (Yes%) | Quality (Yes%) |
>    | ------------------- | :-------------: | :----------: | :-----------: | :------------: |
>    | GPT-4o-2024-11-20   |       100       |      99      |      88       |      100       |
>    | Qwen-Max-2025-01-25 |       100       |     100      |      100      |      100       |
>    | Grok-3              |       100       |     100      |      100      |      100       |
>    | DeepSeek-R1         |       100       |     100      |      100      |      100       |
>    | Human               |       99        |     100      |      94       |       98       |
>
> 4. The baselines in Table 2 are exactly those used in prior work (prompt-based baselines), together with previous state-of-the-art methods (Yo’LLaVA and MyVLM as shown in Table 3). Moreover, both LLaVA (CoT) and Yo’LLaVA (CoT) already incorporate knowledge-graph information in CoT inputs. It is important to emphasize that our graph-prompting mechanism is a component of our proposed method and does not exist in prior personalized MLLMs. Therefore, questioning the fairness of the comparison on the basis of graph prompting is not reasonable; rather, it should be regarded as one of our key contributions.
>
> 5. Thank you for your notice, and we will refer to the ICLR rules.
>
> If we have addressed some of your concerns, please do not hesitate to inform us. We hope to hear from you to further enhance our work.

---

### Official Review · Reviewer_wyZf · 2025-10-31

**Soundness:** 3
**Presentation:** 3
**Contribution:** 3
**Rating:** 6
**Confidence:** 4

**Summary:**

This paper introduces a novel task to perform relational reasoning over personalized concepts using Multimodal Large Language Models (MLLMs). A framework based on soft and/or hard graph prompting is proposed to enhance the relational reasoning capabilities of MLLMs by leveraging knowledge graphs and Chain-of-Thought Question Answering (CoTQA) pairs. The paper also presents a new dataset, ReGraP, which contains images, knowledge graphs, and CoTQA pairs for training, and a benchmark to evaluate the relational reasoning between personalized concepts. Experimental results demonstrate that the proposed ReGraP-LLaVA model effectively learns personalized knowledge and performs relational reasoning, achieving superior performance compared to competitive methods.

**Strengths:**

1. It's a crucial problem to enable MLLMs to perform relational reasoning over multiple personalized concepts.
2. The proposed framework based on soft and/or hard graph prompting is well-designed to enhance the relational reasoning capabilities of MLLMs.
3. The paper develops a data generation pipeline for relational question answering synthesis, and also introduces a new dataset and benchmark named ReGraP, which are valuable resources for future research in this area.
4. The paper is well-written and easy to follow, with clear explanations of the proposed methods and experimental results.

**Weaknesses:**

1. The paper extends the idea of soft/hard prompting beyond previous works (e.g., Yo'LLaVA) by integrating reasoning over knowledge graphs. However, since prompting-based personalization has been explored before, the novelty mainly lies in using structured graph representations and CoT QA data, which could be better emphasized.
2. The paper lacks comparison with several related personalization methods such as RAP-LLaVA, UniCTokens and RePIC. Including or discussing these baselines would strengthen the experimental evaluation.
3. It's unclear how to select between soft and hard graph prompting in different scenarios. The paper should provide more insights or guidelines on when to use each approach for optimal performance.
4. The model needs to be trained on each concepts set, which may limit its applicability in real-world scenarios where personalized concepts may vary widely. The authors should discuss potential strategies to improve the model's generalization to unseen concepts without requiring retraining.
5. Some details should be clarified. For instance, how subgraphs are selected during inference (automatically or predefined). The specific GNN architecture used for graph embedding is also not detailed.

**Questions:**

1. Have the authors compared the performance of directly converting knowledge graphs into textual descriptions (without GNN encoding) versus using soft/hard graph prompting?
2. See weakness.

---

> ### Author Response · Authors · 2025-11-16
> **Rebuttal to Reviewer wyZf**
>
> Dear Reviewer wyZf,
>
> Thank you for your comments and time.  We appreciate the time and effort you've taken to review our work. We apologize for the confusion caused to you, and we hope the following response can clarify your confusion about this paper. Sincerely hope that you will continue to strengthen your support for our work after our rebuttal.
>
> Regarding to weakness:
>
> 1. It is admitted that the idea of soft/hard prompting is one of the key method to enable base models to learn personalized knowledge. Our novelty (contribution) of method lies in the graph representations and CoT QA data that encode the personalized knowledge. We will better emphasize this.
>
> 2. We admitted we do not contain RAP-LLaVA, UniCTokens and RePIC as baseline. RAP_LLaVA is based on retrieval and need to train a retrieval module using a large number of data, while our method is training based and merely relies on LLaVA itself. UniCTokens and RePIC are new works accepted in NIPS this year, and we absolutely do noy have time to make a comparison. However, we admit these works are reference to our works and we will add these reference to our work.
>
> 3. Referring to the Table 5 in section 6.3, the hard prompt has the highest overall accuracy (weighted), while the combination performs better in more cases. This suggests that each method has its own best-performing scenarios following this table.
>
>    | Model                          |  MC (S)   |  MC (D)   |  FIB (S)  |  FIB (D)  |  TF (S)   |  TF (D)   | Desc. (S) | Desc. (D) |
>    | ------------------------------ | :-------: | :-------: | :-------: | :-------: | :-------: | :-------: | :-------: | :-------: |
>    | **ReGraP-LLaVA (Hard)**        | **0.942** |   0.892   |   0.940   |   0.858   | **0.967** |   0.916   |   0.975   | **0.950** |
>    | **ReGraP-LLaVA (Soft)**        |   0.938   |   0.893   |   0.938   |   0.852   | **0.967** |   0.910   |   0.942   |   0.929   |
>    | **ReGraP-LLaVA (Combination)** |   0.929   | **0.898** | **0.943** | **0.860** |   0.960   | **0.917** | **0.983** | **0.950** |
>
> 4. It is admitted that introducing a new set of concepts, features and relations need training a new model. The training-free methods are Retrieval which need to train a retriever and prompting-based methods. A base model never know unseen concepts if there is no relative training data, for example, without training, the base model never knows "The dog in the image is A's dog, and its name is B." Thus, dependence on training can be common in personalization. However, We admit this concern and will investigate in this, for example, the fusion of retrieval and our method can be an approach.
>
> 5. We train the model using pre-defined data (containing subgraphs) and save the weights. During the inference, we load the weights and merge into the base model, and no extra graph-prompting is a provided, all questions are the same for baselines and our models. We load the models and new tokens to the tokenizer,  and directly prompt the questions to the model to generate the answers.
>
>
>    ```python
>    lora_model_path = f'{args.checkpoint_path}/...'
>    model = PeftModel.from_pretrained(model, lora_model_path)
>    model = model.merge_and_unload()  # Merge LoRA into base model
>
>    print("✅ LoRA successfully merged into the base model!")
>
>    sks_token = torch.load(f'{args.checkpoint_path}/...').detach()
>    lm_head = torch.load(f'{args.checkpoint_path}/...').detach()
>    model.get_input_embeddings().weight.requires_grad = False
>    model.lm_head.weight.requires_grad = False
>    model.get_input_embeddings().weight[placeholder_token_ids] = sks_token.to(model.device, dtype=model.dtype)
>    model.lm_head.weight[placeholder_token_ids] = lm_head.detach().to(model.lm_head.weight.device, dtype=model.dtype)
>    print('New tokens are loaded into: ', placeholder_token_ids)
>
>    ......
>
>    num_added_tokens = tokenizer.add_tokens(placeholder_tokens)
>    placeholder_token_ids = tokenizer.convert_tokens_to_ids(placeholder_tokens)
>
>    model.eval()
>    ```
>
> Regarding the question:
>
> 1. Yes. Notably, LLaVA(CoT) serves as an ablated variant of ReGraP-LLaVA without the graph-prompting module, and the performance gains over it in Sections 6.1 and 6.2 prove the effectiveness of graph-promptings. Please refer to the sections for details.
>
>    ### LLaVA (CoT) vs. ReGraP-LLaVA (Ours)
>
>    | Model                             |  MC (S)   |  MC (D)   |  FIB (S)  |  FIB (D)  |  TF (S)   |  TF (D)   | Desc. (S) | Desc. (D) |
>    | --------------------------------- | :-------: | :-------: | :-------: | :-------: | :-------: | :-------: | :-------: | :-------: |
>    | LLaVA (Liu et al., 2023b) **CoT** |   0.885   |   0.829   |   0.890   |   0.817   |   0.947   |   0.877   |   0.917   |   0.867   |
>    | **ReGraP-LLaVA (Ours)**           | **0.942** | **0.892** | **0.940** | **0.858** | **0.967** | **0.916** | **0.975** | **0.950** |

---

### Official Review · Reviewer_tapu · 2025-11-05

**Soundness:** 3
**Presentation:** 3
**Contribution:** 3
**Rating:** 6
**Confidence:** 4

**Summary:**

This paper introduces ReGraP-LLaVA, a multimodal large language model designed to perform relational reasoning over personalized concepts. The authors present a novel dataset (ReGraP) containing 120 sets of personalized knowledge, each with images, knowledge graphs, and Chain-of-Thought question-answering pairs. The method uses both soft prompting (via GNN-based graph embeddings) and hard prompting (via new reasoning tokens) to align knowledge graphs with the model's semantic space. Experiments demonstrate that ReGraP-LLaVA outperforms existing personalized MLLMs on tasks requiring both recognition and multi-step relational reasoning.

**Strengths:**

1. The motivation for this work is clear and compelling. The authors correctly identify that existing personalized MLLMs focus primarily on concept recognition and captioning, while neglecting the relational knowledge and reasoning capabilities that humans naturally employ when understanding personalized contexts.

2. The novelty of the approach is strong. To my knowledge, this is the first work to explicitly construct knowledge graphs for personalized concepts and use them to train MLLMs with relational reasoning capabilities. The dual approach of soft and hard graph prompting provides interesting alternatives for graph-MLLM alignment.

3. The experimental results are strong across multiple task types. ReGraP-LLaVA achieves substantial improvements over both prompt-based and finetuning-based baselines, with particularly impressive gains on difficult reasoning tasks (5.3% over the best finetuning baseline and 8.8% over the best prompt baseline on weighted average).

4. The paper includes useful ablation studies comparing soft vs hard prompting methods, and provides qualitative examples with attention visualizations that demonstrate the model's reasoning process. The human evaluation in Section B adds additional credibility to the results.

**Weaknesses:**

1. The evaluation setup and dataset descriptions are somewhat unclear throughout the paper. While the datasets are described in the main text (Section 5), the tables themselves do not clearly indicate which dataset is being evaluated. For instance, Table 2 does not specify that it evaluates on the ReGraP dataset, while Table 3 evaluates on Yo'LLaVA and MyVLM datasets with different tasks. The authors should add explicit dataset identifiers to table captions and within the tables themselves to improve clarity.

2. Given the strong motivation that prompting alone may not adequately capture personal and relational information, I would expect a more thorough exploration of advanced prompting baselines. The authors make an excellent point that lengthening prompts provide limited context for complex relational reasoning. However, the prompt-based baselines appear to use relatively straightforward description-based prompting. Recent work in visual compositionality has developed more sophisticated prompting approaches that could serve as stronger baselines. For example, Compositional Chain-of-Thought Prompting for Multimodal Large Language Models by Mitra et al. demonstrates how structured CoT prompting can improve compositional reasoning. Similarly, Davidsonian Scene Graph Prompting by Cho et al. shows how scene graph-structured prompts can enhance spatial and relational understanding. Including these or similar baselines would strengthen the evaluation and better demonstrate the necessity of the proposed training-based approach over advanced prompting techniques.

3. The paper would benefit from situating this work within the broader literature on visuolinguistic compositionality and scene graph reasoning. While the authors cite some related work on graphs with MLLMs, there is a rich body of work on compositional visual reasoning, scene graphs for visual understanding, and compositional generalization that seems highly relevant. For instance, work on using scene graphs for visual relationship detection, compositional visual question answering, and structured visual reasoning could provide useful context and potentially inspire additional baseline comparisons or evaluation protocols.

**Questions:**

Most of my feedback is constructive already, describing the perceived weakness and how to resolve it. So please refer to the Weaknesses section.


Can you clarify all the ways in which CoT QA samples are used in your method? From my reading, they appear to serve multiple purposes: (a) as training supervision where questions provide instructions and CoT answers provide target responses, (b) as a "hard-prompt formulation" of knowledge graphs, and (c) as part of your evaluation benchmark. It would be helpful to explicitly enumerate these uses and clarify whether they are also integrated into the graph representation itself in any way beyond being derived from the graph routes.

---

> ### Author Response · Authors · 2025-11-16
> **Rebuttal to Reviewer tapu**
>
> Dear Reviewer tapu,
>
> Thank you for your comments and time.  We appreciate the time and effort you've taken to review our work. We apologize for the confusion caused to you, and we hope the following response can clarify your confusion about this paper. Sincerely hope that you will continue to strengthen your support for our work after our rebuttal.
>
> Regarding to weakness:
>
> 1. Thank you for your notice. The table 2 is the main result and is conducted on the ReGraP. The table 3 shows experiments conducted on datasets provided by the previous works, in order to show the generalization. We will add the details.
>
> 2. We understand that  prompt-based baselines are straight forward and have been developed as strong baselines. These works design structured pipelines for MLLMs reasoning. While designing our experimental settings, we would like to ensure the input of all models are the same, which should be the questions in our benchmark, thus, we utilize the prompting strategies in the previous personalization works (Yo'LLaVA, MyVLM).
>
>    Specifically, regarding the CCoT you have mentioned, to our knowledge, its main contribution is to conduct a **Scene Graph Generation** before prompting the model to generate the answer. However, as the pipeline relies only on the model itself, there is no personalized knowledge injected to the model, therefore it will never learn the personalized knowledge however the inference stage is well-designed. We conducted a few experiments following its orginal settings, and the performance is low and the method is not suitable for personalization in MLLMs.
>
>    | Model                                 |  MC (S)   |  MC (D)   |  TF (S)   |  TF (D)   |
>    | ------------------------------------- | :-------: | :-------: | :-------: | :-------: |
>    | LLaVA-7B (Liu et al., 2023b) (Prompt) |   0.786   |   0.684   |   0.908   |   0.784   |
>    | LLaVA-7B (Liu et al., 2023b) (CCoT)   |   0.559   |   0.361   |   0.645   |   0.532   |
>    | **ReGraP-LLaVA (Ours)**               | **0.942** | **0.892** | **0.967** | **0.916** |
>
>    > As expected, It is obvious that without prompting of the personalized knowledge, the model has a low performance.
>    >
>    > It should be noted that, for example, LLaVA never knows that the yellow hair girl in the image is Nijika who plays the drum as the leader of the band whose name is Kesoku, having xxx relation with xxx, xxx, xxx......., which can be prompted and trained to be injected to the model in other baselines. As the graph goes complex and the question goes difficult, the model seems to guess instead of reasoning.
>
>    On the other hand, we could replace the Scene Graph Generation stage with directly prompting the KG of this set. However, this will absolutely leakage the answer directly for many questions (especially the simple ones), thus is a totally unfair setting.
>
> 3. Thank you for your suggestion. These literature on visuolinguistic compositionality and scene graph reasoning will help us. Due to the page limit, we do not have enough literature review in the main page and we are sorry for that. We will add these literature reviews in the Appendix.
>
> Regarding to the question:
>
> 1. Yes, CoT QA samples have multiple usage. (a) as training supervision where questions provide instructions and CoT answers provide target responses, (b) as a "hard-prompt formulation" of knowledge graphs are correct. Actually, we call textual inputs in the training stage as hard-prompting. During the evaluation, The QA pairs can be not in CoT format. For example, for multiple choice questions, the answers are just A,B, C,D and the model is expected to give a choice.

---

> > ### Comment · Reviewer_tapu · 2025-11-26
> >
> > Thanks to the authors for taking the time to respond to my feedback and questions. I raise my score accordingly.

---

> > > ### Author Response · Authors · 2025-11-26
> > >
> > > Dear Reviewer tapu,
> > >
> > > Thank you for your feedback and recognition. Your efforts truly help improve our work. Accordingly, we will further elaborate the experiments and additional literature review on visuolinguistic compositionality and scene graph reasoning in future versions.
> > >
> > > Best Regards,
> > > Yours, Authors

---

### Author Response · Authors · 2025-11-24
**Kind request for responds from the reviewers**

Dear reviewers,

We would like to express our gratitude to your hard work. We carefully considered all the weakness and questions proposed, and gave feedbacks to address them accordingly.

However, we have not yet received responses from you. With over half of the rebuttal period passed, we kindly request for your responses to us for further discussion, to know if we have adequately addressed the questions and concerns.

We believe that constructive communication with timely responses is essential for the benefit of all parties.

Thank you for your time and assistance.

Best Regards,
The Authors

---

### Author Response · Authors · 2025-12-01
**Summary of the rebuttals**

Dear ACs,

Thank you for your time and hard work. We appreciate the time and effort you've taken to review our work. We are here to summarize the rebuttal.

**To Reviewer tapu**, the reviewer admitted the clear motivation, strong novelty, and sufficient experiments. After the rebuttal, **the reviewer raised the score to 8**, which was before the leakage of data, and could be proved by the comments. Regarding the weakness and questions, the main concerns are more baselines (e.g., CCoT) and more literature review. We have conducted additional experiments to address the issue and due to limitation of pages, we can add more literature review in the appendix.

**To Reviewer wyZf**, the reviewer admitted the merit in motivation, methods and overall quality, and gave a core of 6. The main concerns are the comparison experiments, usage and applicability and details. We have leveraged the experimental results to demonstrate the selection, and add codes (which is open-sourced) to show the details. Besides, we discussed the potential strategies to improve the model's generalization.

**To reviewer DV7e**, the reviewer raised several questions and gave a score of 2. The main concerns are the motivation and data quality. However, the reviewer seems to misunderstand the definition of the task, and have confusion in methods and motivation.

The reviewer doubts the data collection, which is around anime characters and personal items. However, **The personalization is defined as: the unique features, concepts and relations of one's unique item/object/individual...,** thus  collecting data from these **unique anime characters** and **personal items** thus conforms to the scenarios. We have also referred to other works in the same area to demonstrate our statements.

Also, the reviewer doubts the motivation for integrating KG and CoT into personalization. However, these are the methods instead of motivation. The KG and CoT are promising methods to encode the rich personalized relations and features, which can be applied to MLLMs. **The motivation** is to enable the model to learn more personalized knowledge beyond concepts, which is the core idea of previous works, while introducing KGs and CoT is our proposed method to realize it. **The motivation and the method should not be confused**. We also use experimental results to show the effectiveness.

The data quality is doubted because of the usage of LLM. However, LLM pre-annotation and human verification is a common and important data generation method. Also, we additionally conduct human evaluation on the CoT pairs (see the section **Appendix C** in paper for details). We also perform human evaluation on model responses (see the section **Appendix B** in paper for details). These combination further demonstrate that our model better aligns with human preferences and support the effectiveness of our benchmark.

Additionally, we address other issues accordingly and please refer to comment details. We hope to hear from the reviewer while receive no response until the leakage happens. We sincerely hope that ACs can consider our rebuttal when judging our work.

**To reviewer FnoX**, the reviewer admitted the integrity of our work and methods. With a few concerns, the reviewer gave a score of 6. The main concerns are CoT prompting settings for baselines and common problems of finetuning (e.g., catastrophic forgetting). We have ensured the fairness in comparisons and more CoT based methods may refer to CCoT mentioned before. We also show experimental results accordingly.

**To reviewer VFmz**, the reviewer admitted the detailed and comprehensive research content, raised concerns in details and few comparison datasets. The reviewer also ask for ablation studies. However, the ablation studies requested by the reviewer are just the ablation studies we have already **presented in the Appendix A Additional Ablation Studies**. Also, to our knowledge,  by the end of the paper, in the personalization of LLMs, there were only these two well-documented and high-cited datasets, which were already published. We addressed the issues accordingly. However, we raise concerns that, as the reviewer ask for experiments that just presented in the paper, does he/she really read our work carefully before making the review? Also, we believe that, as we have already conducted the experiments and address issues, the score should be raised if the reviewer carefully read the article.

---

### Author Response · Authors · 2025-12-01
**Summary of the our contributions**

Dear ACs,

Thank you for your time and hard work. We appreciate the time and effort you've taken to review our work. We are here to summarize the main contributions of our work.

- **ReGraP Dataset & Pipeline.** We build **ReGraP**, a dataset and data-generation pipeline for personalized MLLMs that jointly uses images, **personalized knowledge graphs (KGs)**, and **Chain-of-Thought (CoT) QA pairs**. The pipeline constructs KGs over personalized concepts and then generates CoT QA pairs grounded in these graphs.

- **ReGraP-LLaVA Training framework.** We propose **ReGraP-LLaVA**, a personalized MLLM that integrates KGs via **soft prompts** (GNN-based KG embeddings) and **hard prompts** (natural language relational descriptions with new entity/relation tokens), together with CoT QA supervision. This enables the model to not only recognize personalized concepts, but also **use their relational structure to perform multi-step reasoning** for personalized image understanding and QA.  **To our knowledge, we are the first to emphasize the importance of relations and features of objects in personalized MLLMs.**

- **ReGraP Benchmark.** We introduce the **ReGraP benchmark**, covering Multiple-Choice, Fill-in-the-Blank, True/False, and Descriptive questions under both open- and closed-ended settings. The benchmark is designed to scale in difficulty, explicitly evaluating both **personalized knowledge acquisition** and **relational reasoning / knowledge-connection capabilities** of MLLMs.

---

### Meta-Review · Area_Chair_CQQe · 2026-01-07

**Summary:**

Reviewers acknowledged the strong motivation and novelty of jointly modeling personalized concepts, relations, and multi-step reasoning, a gap in existing work. However, concerns were raised regarding dataset scope, data generation reliability, fairness of baselines, missing comparisons to recent personalization methods, and clarity on prompt selection guidelines. The authors provided detailed rebuttals, clarifying the alignment of their data collection with personalization definitions, presenting human evaluations validating data quality, adding ablation results, explaining baseline choices. However, two reviewers still did not believe that the paper could be accepted.

**Reviewer Concerns:**

The authors addressed every weakness raised.

**Reviewer Scores:**

Reviewer tapu (initial 6) explicitly stated in their post-rebuttal comment that they may raise their score to 7 or 8.
Reviewer wyZf (initial 6) did not provide any follow-up response after the rebuttal, so their score is assumed to remain 6.
Reviewer DV7e (initial 2) did not engage further after the authors’ response and gave no indication of revising their assessment; their score is therefore kept at 2.
Reviewer FnoX (initial 6) asked clarifying questions that were addressed by the authors, but did not indicate any change in their evaluation; their score is maintained at 6.
Reviewer VFmz (initial 4) raised concerns that the authors clarified were already covered in the submission, but did not respond afterward or signal a score adjustment; thus, their score remains 4.

---

### Decision · Program_Chairs · 2026-01-26

Reject